# Characterizing the Dynamic Protein and Amino Acid Deposition in Tissues of Pregnant Gilts: Implications for Stage-Specific Nutritional Strategies

**DOI:** 10.3390/ani15142126

**Published:** 2025-07-18

**Authors:** Christian D. Ramirez-Camba, Pedro E. Urriola, Crystal L. Levesque

**Affiliations:** 1Department of Animal Science, University of Minnesota, St. Paul, MN 55108, USA; ramir643@umn.edu; 2Department of Animal Science, South Dakota State University, Brookings, SD 57007, USA

**Keywords:** gestation model, protein deposition, amino acid requirements, swine nutrition, reproductive performance, dietary optimization, gilt growth

## Abstract

Ensuring that pregnant sows receive proper nutrition is crucial for their health and the development of their piglets. Current gestation models assume that protein and amino acids are deposited at a constant rate, but experimental data show that deposition rates vary across different stages. By developing a new model that accounts for changes in tissue growth and composition, we found that protein deposition fluctuates significantly throughout gestation. Our findings suggest that sows may require different diets at different times, particularly more protein and amino acids early in pregnancy than previously thought. Additionally, we identified gaps in understanding protein deposition in both the sow and conceptus, which, by studying further, could lead to improved feeding strategies that better support maternal and fetal development. By advancing knowledge of protein deposition dynamics, our research lays the foundation for precision feeding strategies that enhance pig health, reproductive outcomes, and farm efficiency.

## 1. Introduction

Understanding the dynamics of crude protein (CP) and amino acid (AA) deposition in pregnant sows is essential for optimizing nutritional management and improving reproductive performance. Previous models, such as the NRC (2012) gestating sow model [1], provide a strong foundation for understanding CP and AA deposition. However, there is significant potential to enhance these models by incorporating greater physiological complexity. For example, the NRC (2012) [1] gestating sow model assumes a constant proportional CP and AA content in sow tissues throughout gestation. However, empirical data show that the weight, CP and AA content of conception products vary as gestation progresses [2,3,4]. Integrating these dynamic changes into modeling approaches could uncover new complexities and highlight gaps in our understanding of the biological processes involved.

This study aims to build on the NRC (2012) [1] gestating sow model by developing a mechanistic framework that adheres to its broad principles, while incorporating variations in gestational tissue growth and nutritional demands to enhance our understanding. Unlike traditional gestation models, this study focuses on characterizing previously unexplored growth and development processes during gestation, rather than developing a predictive model for AA requirements. By highlighting knowledge gaps, the study aims to contribute to a more precise understanding of the potential nutritional needs of gestating gilts, paving the way for improved dietary formulations and management practices.

## 2. Materials and Methods

The gestational model presented herein outlines the dynamic partitioning of body weight, as well as CP and AA deposition, across various tissues constituting the gilt throughout the gestational period. The model includes the following key tissues: allantoic fluid, amniotic fluid, placenta, uterus, fetus, mammary gland, and maternal body. These tissues were chosen because they are the primary tissues in a pregnant sow and align with those selected in the gestating sow model by the NRC (2012) [1]. This model was developed in three steps. Firstly, data on wet weights and CP content of the different tissues were extracted from published peer-reviewed articles through a systematized literature search. Additionally, data on daily whole-body CP retention were collected in this initial step. In the second step, wet growth and CP content curves were formulated based on the data gathered in step 1 for the allantoic fluid, amniotic fluid, placenta, uterus, fetus, and mammary gland. Additionally, in this second step, the daily whole-body CP retention curve was developed. Finally, in the third step, the developed curves were integrated into an algorithm to estimate the wet growth and CP retention of the pregnant gilt. This algorithm was designed to use as inputs the body weight at breeding, litter size, average piglet birth weight, and number of available teats.

### 2.1. Step 1: Data Collection

To obtain the data regarding the wet growth, CP and AA content of the allantoic fluid, amniotic fluid, placenta, uterus, fetus, and mammary gland, a systematized literature search was performed. The systematized search was informed by the PRISMA guidelines [5] and conducted in March 2024 in three electronic databases—Web of Science, PubMed, and Scopus. We conducted our search using the keywords “gilt” and either “amniotic”, “allantoic”, “placenta”, “uterus”, “fetus”, or “mammary”, combined with “pregnant” or “gestation” and “composition.” The search was restricted to articles published in English.

Eligible studies for the growth and CP deposition models were those that reported wet weights or CP contents at a minimum of three different time points throughout gestation. The selected studies from the eligible pool were those that reported the most time points for wet weight and CP content, and especially those that covered the longest duration of the gestational period—this was considered the strongest criterion in the selection process. This approach prioritized broader data coverage across gestation, as it allowed for a more accurate and continuous representation of temporal growth and deposition patterns. When two studies reported a similar number of time points, the one with broader gestational coverage was selected. Therefore, for each tissue, a single study was chosen based on the Best Evidence Synthesis (BES) principle, which emphasizes using the best available dataset without aggregating all available studies [6,7,8]. The BES approach was employed due to large variability between studies, including differences in reproductive performance, genetics, and the studied timeframe during gestation. Thus, the initial search yielded 754 studies, of which after careful review, 19 were considered eligible (Appendix A) and 5 were selected (Table 1).

To obtain the data regarding the whole-body protein retention, a second systematized literature search was conducted in March 2024 using the same three electronic databases—Web of Science, PubMed, and Scopus. This search strategy involved using the keywords “gilt” and “nitrogen” and “balance.” The search was again limited to articles published in English. Studies were considered eligible if they reported N retention estimates at a minimum of three different time points throughout gestation. Following the BES principle, the selected study was chosen from the eligible pool based on the number of time points and the duration of gestational coverage, with preference given to studies offering broader data coverage when both criteria were similar. The initial search yielded 86 studies, of which after careful review, 12 were considered eligible (Appendix A), and the study performed by Miller et al. [9] was selected (Table 2).

From the systematized literature search, eligible studies for the AA deposition model were those that reported AA composition of the selected tissues. A total of 9 studies were considered eligible (Appendix A). From the eligible pool of studies, following the BES principle, 3 studies were selected for the development of essential AA deposition curves (Table 3). The selected studies for each tissue were those from the eligible pool that included the most time points and reported all essential amino acids. When studies had a similar number of time points, preference was given to those with broader gestational coverage. Preference was given to studies that reported essential AA deposition in relation to tissue CP concentration as these could be directly linked to the daily CP deposition model.

The systematized search provided information on the concentration of essential AA in amniotic and allantoic fluids, placenta, uterus, and fetus. However, no data were found for the mammary gland. Therefore, an additional systematized search was conducted in March 2024 in the same three databases: Web of Science, PubMed, and Scopus. The goal of this second search was to obtain the AA profile of the mammary gland during lactation and extrapolate this information to the AA profile of the mammary gland during gestation. We conducted our search using the keywords “mammary gland” and “composition”, combined with “amino acid”, “lactation”, and either “gilt” or “sow.” The search was restricted to articles published in English. This second search identified 24 unique citations. After reviewing all of them, only one study was found to report the AA profile of the mammary gland tissue during lactation. The study performed by Kim et al. [10] reported the AA composition of the mammary gland at days 5, 10, 14, 21, and 28 of lactation. The AA profile of the mammary gland at day 5 of lactation was chosen under the assumption that it would more closely reflect the AA profile of the mammary gland during gestation.

### 2.2. Step 2: Curve Development

Using the collected data, mathematical functions were developed to calculate wet weight, CP and AA content throughout gestation for each of the targeted tissues. CurveExpert Professional software (version 2.7.3) was employed for this purpose [11]. The curve development process involved the combination of various equations encompassing a wide range of families such as exponential, power, growth, dose–response, sigmoidal, and distribution family models. The data obtained from the selected papers provided multiple time points throughout gestation; however, no single dataset was 100% complete, so educated guesses about the possible trend of growth, CP and AA composition were formed. For example, data points from a few weeks after breeding and several days before farrowing were often missing. In this case, the data trend was extended.

**Table 1 animals-15-02126-t001:** Considered studies for developing growth and protein deposition curves for the key tissues in the model.

Tissue/Studies	Number of Time Points		Gestation Period, Days	Selected
	Wet Weight, kg	Crude Protein, % of Wet Weight		
Allantoic fluid				
Knight et al. (1977) [12]	11	11	20–100	✔
Tarraf and Knight (1995) [13]	4	NR	40–100	
Wu et al. (2017) [14]	7	NR	20–114	
Amniotic fluid				
Knight et al. (1977) [12]	9	9	30–100	✔
Wu et al. (2017) [14]	7	NR	20–114	
Placenta				
Guimarães et al. (2014) [15]	3	NR	50–106	
Jang et al. (2017) [3]	6	6	43–108	
Knight et al. (1977) [12]	11	NR	20–100	
McPherson et al. (2004) [16]	NR	6	45–110	
Montes et al. (2018) [17]	4	NR	30–90	
Tarraf and Knight (1995) [13]	4	NR	40–100	
Vallet and Freking (2006) [18]	4	NR	45–105	
Vallet et al. (2010) [19]	5	NR	25–105	
Wise et al. (1997) [20]	3	NR	30–104	
Wright et al. (2016) [21]	5	NR	22–42	
Wu et al. (2017) [14]	7	7	20–114	✔
Uterus				
Jang et al. (2017) [3]	6	6	43–108	✔
Knight et al. (1977) [12]	11	NR	20–100	
Fetus				
Finch et al. (2002) [22]	3	NR	30–114	
Guimarães et al. (2014) [15]	3	NR	50–106	
Jang et al. (2017) [3]	6	6	43–108	
Ji et al. (2005) [2]	6	NR	45–112	
Knight et al. (1977) [12]	11	NR	20–100	
McPherson et al. (2004) [16]	6	6	45–110	
Montes et al. (2018) [17]	4	NR	30–90	
Tarraf and Knight (1995) [13]	4	NR	40–100	
Vallet and Freking (2006) [18]	4	NR	45–105	
Vallet and Freking (2007) [23]	4	NR	45–105	
Vallet et al. (2010) [19]	5	NR	25–105	
Wise et al. (1997) [20]	3	NR	30–104	
Wright et al. (2016) [21]	5	NR	22–42	
Wu et al. (1999) [4]	5	5	40–114	✔
Wu et al. (2017) [14]	7	NR	20–114	
Mammary gland				
Ji et al. (2006) [24]	6	6	45–112	✔
Sørensen et al. (2002) [25]	12	NR	10–114	

NR = Not Reported. Studies marked with a ✔ were selected for the development of growth and protein deposition curves.

**Table 2 animals-15-02126-t002:** Studies considered for developing the whole-body protein retention curve in pregnant gilts.

Studies	Number of Time Points	Gestation Period, Days	Selected
Clowes et al. (2003) [26]	3	33–105	
Corley et al. (1983) [27]	5	90–112	
Dunn and Speer (1991) [28]	6	45–99	
Jones and Maxwell (1982) [29]	3	30–90	
King and Brown (1993) [30]	3	30–93	
Miller et al. (2016) [9]	5	42–112	✔
Miller et al. (2018) [31]	4	49–108	
Navales et al. (2019) [32]	3	41–107	
Noblet and Etienne (1987) [33]	3	60–110	
Ramirez-Camba et al. (2020) [34]	3	41–107	
Willis and Maxwell (1984) [35]	3	30–90	
Yang et al. (2021) [36]	3	35–98	

Studies marked with a ✔ were selected for the development of the whole-body protein retention curve.

**Table 3 animals-15-02126-t003:** Studies considered for estimating the amino acid (AA) composition of different tissues in the pregnant gilt across gestation. The symbol (×) indicates the form in which essential AA composition data was available for extraction, and the symbol (✔) denotes the selected studies for AA composition concentration estimation.

Tissue/Studies	Number of Time Points	Wet Basis	Crude Protein Basis	Selected
Allantoic fluid				
Li et al. (2014) [37]	1	×	NR	
Li et al. (2023) [38]	1	×	NR	
Wu et al. (1998) [39]	2	×	NR	✔
Amniotic fluid				
Li et al. (2014) [37]	1	×	NR	
Wu et al. (1998) [39]	2	×	NR	✔
Li et al. (2023) [38]	1	×	NR	
Placenta				
Jang et al. (2017) [3]	6	×	×	
Wu et al. (2017) [14]	7	NR	×	✔
Self et al. (2004) [40] ^1^	9	×	NR	
Li et al. (2014) [37]	1	×	NR	
Wu et al. (1998) [41]	2	×	NR	
Uterus				
Jang et al. (2017) [3]	6	×	×	✔
Fetus				
Jang et al. (2017) [3]	6	×	×	✔
Wu et al. (1999) [4]	5	×	×	
Wu et al. (2017) [14]	7	×	NR	
Everts and Dekker (1995) [42]	1	×	×	

^1^ Only branched-chain amino acids were reported. NR = Not Reported.

The wet weight curves for amniotic and allantoic fluids, placenta, uterus, and fetus are expressed as grams of wet tissue per fetal pig at each day of gestation. The wet weight curve for the mammary gland is calculated as grams of wet tissue per individual gland at each day of gestation. The CP content curves for amniotic and allantoic fluids, placenta, uterus, fetus, and mammary gland are represented as a percentage of wet weight. To determine the CP content per fetal pig in the products of conception and mammary gland, the wet weight curves were multiplied by the CP content curves for each tissue. The developed pattern of daily weight and CP content of all tissues considered are shown in Figure 1 and the respective equations are reported in Table 4.

The daily CP deposition in the whole body of the pregnant gilt was estimated based on Miller et al. [9], leading to the formulation of Equation (19) (Figure 2A). Subsequently, the daily CP deposition was cumulatively added, resulting in Equation (20) (Figure 2B). It should be noted that, as mathematically described in Equation (20), the cumulative sum of a function is analogous to its integral.

The AA concentration curves were calculated for arginine (Arg), cysteine (Cys), histidine (His), isoleucine (Ile), leucine (Leu), lysine (Lys), methionine (Met), phenylalanine (Phe), threonine (Thr), tryptophan (Trp), and valine (Val). The AA concentrations in allantoic and amniotic fluids were calculated from the results of Wu et al. [39]. The authors provided AA concentrations at days 40 and 60 of gestation, which were averaged to represent a single AA concentration at day 50. Essential AA levels in these fluids were determined from animals fed a 13% CP diet and are expressed as grams of AA per kg of fluid. The concentrations in the allantoic fluid are Lys (0.25291), Arg (0.41420), His (0.05768), Ile (0.01108), Leu (0.01174), Met (0.00642), Cys (0.02423), Phe (0.00764), Thr (0.09979), Trp (0.01317), and Val (0.03028). For the amniotic fluid, the concentrations are Lys (0.04872), Arg (0.03567), His (0.00850), Ile (0.01305), Leu (0.01220), Met (0.00761), Cys (0.00448), Phe (0.00797), Thr (0.01540), Trp (0.00368), and Val (0.02542).

For the placenta, essential AA concentrations were determined based on findings reported by Wu et al. [14] (Figure 3A). For the uterus and fetus, AA concentrations were based on Jang et al. [3] (Figure 3B,C, respectively). Essential AA concentrations in the placenta, uterus, and fetus were fitted using a quadratic function Equation (21), with results expressed as grams of AA per 100 g of tissue CP. The coefficients from Table 5 can be applied to Equation (21) to model the deposition of each essential AA, as illustrated in Figure 3.

Concentrations of AAs in the mammary gland, according to Kim et al. [10], are as follows (grams per 100 g of mammary gland CP): Lys (7.44), Arg (6.23), His (2.47), Ile (4.02), Leu (8.24), Met (1.97), Cys (1.57), Phe (4.33), Thr (4.29), Trp (1.20), and Val (5.59). The estimation of the AA profile deposited in the maternal tissue gain may be challenging, as different tissues grow during gestation such as soft and connective tissue, organs, skin, hair, and blood, each of which have a different AA profile. However, for the current modeling approach, the AA profile of the maternal body gain during gestation of gilts was assumed to be similar to the AA concentration in the whole body of 146 kg pigs reported by Mahan and Shields, Jr [43]. The concentrations of AAs in the maternal body, calculated as grams of AA per 100 g of CP, are Lys (7.4), Arg (5.9), His (3.2), Ile (3.9), Leu (7.2), Met (1.8), Cys (1.3), Phe (4.0), Thr (3.7), Trp (1.3), and Val (5.0).

### 2.3. Step 3: Model Development

#### 2.3.1. Growth Model

The subsequent step involved the computation of daily CP content in the products of conception and the mammary gland within the entire body of the pregnant gilt as shown in Equation (22). This entailed multiplying the CP content in each component of the products of conception (amniotic and allantoic fluids, placenta, uterus, and fetus) by the litter size, a key model input. Additionally, a specific correction was applied to the fetus by multiplying the daily fetus weight (as per Equation (15)) by the input piglet birth weight in grams divided by 1460.23, which represented the calculated fetus weight (g) at day 115 of gestation as shown in Equation (13). This correction ensures proportional adjustments to the daily weight and CP content if the user inputs a different average piglet birth weight. Subsequently, the CP content in the mammary gland was multiplied by the number of available teats, another crucial input in the model. Thus, the final computation for the daily CP content in the products of conception and mammary gland was carried out in accordance with Equation (22).

Subsequently, the cumulative wet weight gained by the maternal body throughout gestation was calculated by subtracting the whole-body CP content as shown in Equation (20) from the CP content in the products of conception as shown in Equation (22) and then dividing the result by the CP content in the maternal body, set at 22% as shown in Equation (23). This 22% corresponds to the CP content in the maternal body, encompassing all tissues including soft and connective tissue, organs, skin, hair, and blood. Initial calculations using the CP content of gilt carcasses (approximately 16%) in Equation (23) yielded overestimated maternal wet weights. To address this, a CP content value of 22% for maternal body weight gain was adopted to align with empirically observed maternal body weights. Although this revised value may seem high compared to carcass CP of growing pigs and sows, it remains within the range of the percentage CP in lean muscle [44,45]. Appendix A presents a diagram illustrating how the growth model integrates Equations (1)–(22) with the objective of estimating the maternal body wet weight as shown in Equation (23).

After estimating the wet weights of the products of conception and the mammary gland throughout gestation, the final step involved combining these estimates with the model inputs, the calculation of maternal body wet weight as shown in Equation (23), and the initial body weight, as outlined in Equation (24). Appendix A presents a diagram illustrating how the growth model integrates the calculated wet weights of the products of conception and the mammary gland with the calculated maternal body weight.

#### 2.3.2. Daily CP Deposition Model

For the growth model, equations were developed to describe the CP content of each key tissue that comprises the animal. These CP content curves represent the cumulative CP deposition throughout gestation. For example, Equation (3) (Figure 1C) shows the CP content of amniotic fluid per fetal pig. By day 70, this reaches approximately 4 g, indicating that over the first 70 days, the amniotic fluid has accumulated around 4 g of CP per fetal pig. For the daily CP deposition model, cumulative CP deposition estimates were converted into daily deposition values using the discrete derivative. Mathematically, the discrete derivative of Equation (3) (Δ Equation (3)) is calculated asΔ Equation (3) (day) = Equation (3) (day + 1) − Equation (3) (day), for day = 1 to 114

Thus, the daily amniotic fluid CP deposition (g·fetal pig^–1^·day^–1^) was calculated according to Equation (25) (Table 6). Following a similar methodology, the daily CP deposition was calculated for the allantoic fluid, placenta, uterus, fetus, and mammary gland using Equations (26)–(30), respectively (Table 6). For daily CP deposition in maternal tissue, the cumulative maternal body weight, calculated using Equation (23) from the growth model (Table 4), was converted to cumulative maternal body CP by multiplying it by 22% (the CP content in maternal body gain), as shown in Equation (31). The discrete derivative was then applied to Equation (31) to estimate daily maternal body CP deposition, resulting in Equation (32).

The next step involved calculating the daily CP deposition in the products of conception and the mammary gland considering the inputs of the model. Thus, by multiplying the daily CP amniotic fluid deposition per fetal pig by the litter size, the daily amniotic fluid CP deposition in g per day was calculated as shown in Equation (33). Similar calculations were performed for the allantoic fluid, placenta, and uterus, as shown in Equations (34)–(36), respectively. In the case of estimating the daily fetus CP deposition, an additional correction was applied by multiplying the daily fetus CP deposition per fetal pig by the litter size and dividing by 1460.23 as shown in Equation (37). This correction ensures proportional adjustments to the daily CP content if the user inputs a different average piglet birth weight than that used for the development of the fetal growth and CP equations. Equation (38) shows the daily CP deposition in the mammary gland by multiplying its daily CP deposition as shown in Equation (30) by the number of available teats.

Equations (32)–(38) describe the daily CP deposition in each tissue of the whole body. By summing Equations (32)–(38), the total CP retention equals the whole-body CP retention calculated according to Equation (19). As a result, the daily CP deposition model dynamically partitions the retained protein into the different tissues of the pregnant gilt based on the inputs of the model. Appendix A presents a diagram illustrating how the CP deposition model integrates all the necessary equations.

#### 2.3.3. Daily Essential AA Deposition Model

The essential AA concentration in the allantoic and amniotic fluids was calculated as a proportion of tissue wet weight. Equations (39) and (40) show the daily wet weight deposition for amniotic and allantoic fluids, respectively, per fetal pig (Table 7). To calculate the essential AA deposition in the amniotic and allantoic fluids, Equations (39) and (40) were multiplied by the model input litter size and their respective AA concentration in the tissue, resulting in Equations (41) and (42). The AA concentrations were divided by 1000, as they were reported on a per-thousand basis (g/kg).

The daily essential AA concentrations in the placenta, uterus, fetus, mammary gland, and maternal body were calculated as a proportion of CP in the tissues. To calculate the daily essential AA deposition in the placenta, the daily placenta CP deposition per fetal pig as shown in Equation (27) was multiplied by the model input litter size and the respective essential AA concentration in the tissue, resulting in Equation (43). The placental AA concentrations were calculated using Equation (21) with the coefficients shown in Table 5. The AA concentrations were divided by 100, as they were calculated on a percentage basis (g/100 g). Following the same logic used for calculating the daily essential AA deposition in the placenta, the essential AA deposition in the uterus and fetus was calculated according to Equations (44) and (45), respectively. To calculate the daily essential AA deposition in the mammary gland, Equation (30) was multiplied by the model input number of available teats, and their respective AA concentration in the tissue, resulting in Equation (46). To calculate the daily essential AA deposition in the maternal body, the daily maternal CP deposition as shown in Equation (32) was multiplied by their respective AA concentration in the tissue, resulting in Equation (47). Finally, to calculate the whole-body essential AA deposition, Equations (41)–(47) were summed as shown in Equation (48). Appendix A provides a diagram that demonstrates how the AA deposition model incorporates all the required equations.

## 3. Results

The growth, CP, and AA deposition curves, and consequently the entire model, were developed using the BES principle, which prioritized the highest-quality evidence over an exhaustive inclusion of all available data [6,8]. This approach was chosen due to variability in the available data and the complexity of pregnancy-related tissue growth. For example, Figure 4 shows Lys concentrations in the placenta across gestation. Two studies identified in the literature reported data from more than three gestational time points, but their findings differed significantly: Jang et al. (2017) [3] reported Lys concentrations nearing 8 mg/g tissue, while Wu et al. (2017) [14] reported values up to 4 mg/g tissue. As shown in Figure 4, a simple average of these values may not accurately reflect the progressive deposition of Lys in the placenta. Due to the significant differences among the reported values, the BES principle was applied. This ensured that tissue growth and development were modeled using the highest-quality dataset, rather than aggregating conflicting data. While this approach acknowledges potential experimental errors in individual studies, it minimizes the risk of extrapolation errors from disparate sources, which were considered a more substantial source of error. In addition, the objective of the current model was not to estimate nutritional requirements, as previous mechanistic models were designed to do, but rather to identify knowledge gaps. In this context, we believe that applying the BES principle is appropriate. Therefore, the results should be viewed not as a definitive standard for growth, CP, and AA deposition but as a partial snapshot of the composition of pregnant sows with the intent to identify areas that require further understanding.

To encourage broader use and facilitate further improvements, we have made the R code used to develop the current model available through an open-source framework. The Appendix A contains the code for the algorithms and equations implemented in the model, enabling other researchers to refine, adapt, and expand upon the work as new data and insights emerge. While the BES principle was applied in the current study and a single dataset was selected for each tissue, the compiled data on wet weights across gestation for allantoic fluid, amniotic fluid, placenta, uterus, fetus, and mammary gland are shown in Appendix A, respectively. Furthermore, Appendix A display the compiled data for CP composition across gestation for the placenta and fetus, respectively. Appendix A shows the compiled data on whole-body protein deposition across gestation.

### 3.1. Growth Model

The growth model was constructed based on data from six scientific articles published between 1977 and 2017 (Table 1 and Table 2), yielding 23 initial equations (Table 4). The wet weights and CP contents of each tissue were extracted from a single study to minimize data variability. Although the model incorporates data from multiple sources along with a set of assumptions, our methodology involved evaluating the model outputs using empirical data that were not employed during the model development. This evaluation was conducted to validate the model’s accuracy in representing animal responses and is described in detail in the next section.

#### 3.1.1. Growth Model Evaluation

To test the accuracy of the model in describing the wet growth of the pregnant gilt, data from three studies were employed to fit the growth data from the model (Buis [46]; Thomas et al. [47]; and Stewart [48]). The model demonstrated high goodness of fit with R^2^ values of 0.99, 0.98, and 0.99, respectively (Figure 5). However, the calculated whole-body weight during the entire gestation period deviated from the empirically observed values. The predicted whole-body weight gain during gestation for Buis [46], Thomas et al. [47], and Stewart [48] was 73.1, 76.8, and 72.8 kg, respectively. In contrast, empirically reported values are approximately 62 kg [1].

Thus, although R^2^ values indicate a strong statistical fit, goodness-of-fit metrics were not the primary evaluation criterion for the model. Instead, the main emphasis was placed on biological relevance, that is, whether the predicted outputs aligned with known physiological patterns in pregnant gilts. This biologically grounded evaluation was considered more appropriate given the model’s exploratory purpose and the use of aggregated data (Figure 5), which can artificially inflate statistical fit.

Therefore, the model’s prediction of weight gain during gestation was considered to be overestimated. This discrepancy was attributed to weight loss during the initial days of gestation, which the developed model did not consider. This body weight loss was observed by Buis [46] and attributed to gilts learning to eat from the electronic feeders. Additionally, in Thomas et al. [47], the initial 5-day body weights were excluded due to potential inaccuracies stemming from scale calibration issues, with an average body weight of approximately 155 kg recorded on day 6. However, Thomas [49] reported an initial body weight of 165 kg for the same group of animals. Hence, Thomas’ [49] data also suggest an initial decrease in body weight during early pregnancy. A similar weight loss that is shown in Buis [46] and Thomas et al. [47] can be observed in the data reported by Stewart [48].

The data reported by Buis [46], Thomas et al. [47], and Stewart [48] share the commonality that all gilts were fed using electronic feeders and were housed in pens, suggesting that an adaptation to the environment and the new group of animals may explain the reduction in body weight during the first 2 weeks of gestation. However, in the experiment conducted by Ji et al. [2], where gilts were allocated in stalls, a similar phenomenon appears to have occurred (Figure 6). Although Ji et al. [2] used the slaughter technique, and the weights shown in Figure 6 are from a different group of animals, their data also suggest a small reduction in body weight during the initial days of gestation. This intriguing consistency across different studies warrants further investigation to fully understand its underlying causes.

#### 3.1.2. Growth Model Adjustments to Fit Empirical Data (Model Calibration)

To address the weight gain overestimation during early gestation, the growth model was adjusted by modifying the daily whole-body CP retention curve. A calculated protein deficit (Equation (49); Figure 7A) was subtracted from the original CP retention as shown in Equation (19) to yield new daily CP retention estimates (Equation (50); Figure 7B). With these adjustments, the updated model estimated whole-body weight gains of 65.1 kg, 66.3 kg, and 65.1 kg for the Buis (2016) [46], Thomas et al. (2018) [47], and Stewart (2020) [48] datasets, respectively, aligning more closely with empirical values (Figure 8).(49)$$\mathrm{Potential}\;\mathrm{protein}\;\mathrm{deficit},\;g/d=\frac{299.86\cdot {\mathrm{day}}^{-7.2056}}{8.3295^{-7.2056}+{\mathrm{day}}^{-7.2056}}$$(50)$$\mathrm{Daily}\;\mathrm{whole-body}\;\mathrm{CP}\;\mathrm{deposition},\;g\cdot {\mathrm{day}}^{-1}=\mathrm{Equation}\;(19)-\mathrm{Equation}\;(49)$$

### 3.2. Daily CP Deposition Model

To explore the novel perspectives on animal responses offered by the daily CP deposition model, our discussion employs a comparative analysis against the well-regarded NRC (2012) [1] gestating sow model. This comparison is warranted because our daily CP deposition model builds on the framework established by the NRC (2012) [1]. Highlighting the differences between the two models facilitates the identification of new insights into CP deposition during gestation.

Figure 9A presents a visual representation of the growth model by tissue, while Figure 9B illustrates the daily CP deposition model by tissue with the model parameterized to represent the same gilt. It is important to mention that both the growth model and the CP deposition model are interconnected; when the inputs of the model are changed, both models vary simultaneously. The growth model is adjusted so that its tissues contain the amount of CP that matches the levels of whole-body CP retention. Our models’ ability to calculate growth and CP deposition in an interconnected manner differs from previous models like the NRC (2012) [1]. The NRC (2012) [1] gestating sow model equations estimate CP deposition for the products of conception and the maternal body independently. This feature of our models has the potential to help study the competition between tissues for nutrients during gestation.

Figure 10 illustrates the comparison between our proposed CP deposition model and the NRC (2012) [1] gestation CP deposition model. The daily whole-body protein retention predicted by our CP deposition model (Figure 10A) is higher than the estimates provided by the NRC (2012) [1] model (Figure 10B). This difference may be attributed to the data on which the models were developed. The NRC (2012) [1] CP deposition model was based on studies reporting nitrogen retention during gestation from 1965 to 2003, whereas our model relies on nitrogen balance data reported by Miller et al. (2016) [9]. Consequently, the CP deposition predictions from the NRC (2012) [1] model reflect the genetics of pregnant sows from over two decades ago, while the estimates based on Miller et al. [9] may more accurately represent current animal genetics. Furthermore, our model shows increased daily CP retention not only in the whole body but also in the fetus and uterus compared to the NRC (2012) [1] model. These observations suggest that the NRC (2012) [1] gestating sow model may underestimate CP retention.

### 3.3. Daily AA Deposition Model

Figure 11 shows the daily essential AA deposition in the whole body of gilts throughout gestation. Based on observations from the growth model, the AA deposition predictions for early gestation are considered to be underestimated and should be interpreted with caution. The model predictions indicate that the AA profile of the deposited protein remains relatively constant during early and mid-gestation but varies during late gestation, particularly in the essential AA ratios relative to Lys. This change in the essential AA profile and AA ratios during late gestation may be explained by the changes in AA composition that occur in the fetal pig. As shown in Figure 3C, the concentration of essential AAs in fetal protein decreases as pregnancy progresses, with the exception of Arg, which increases. This shows that as the fetal pig grows, the proportion of Lys in fetal protein progressively decreases, accompanied by a proportional increase in the demand for Arg and/or non-essential AAs for fetal protein synthesis during late gestation.

The AA concentrations in fetal protein used in this study are based on Jang et al. [3]. However, data from Wu et al. [4], which also reported AA concentrations in relation to fetal CP content (Table 3), show a similar trend in the proportional reduction in all essential AA concentration except Arg. Thus, our study shows an increased proportional deposition of Arg and non-essential AAs for fetal protein synthesis during late gestation compared to Lys. These findings indicate that the AA profile of the deposited protein differs during late gestation.

The results of the daily essential AA deposition model predict that Lys is the AA deposited in the greatest quantities during early and mid-gestation in gilts but becomes the third most deposited essential AA during late gestation. Additionally, daily Lys deposition remains relatively constant from day 80 to farrowing, ranging from 9.4 to 9.6 g/d, reflecting only a 3.6% increase. In contrast, the deposition of other essential AAs shows greater incremental changes from day 80 to farrowing: Arg (+31.31%), Cys (+28.97%), Thr (+18.64%), Leu (+17.05%), Val (+14.51%), Phe (+13.63%), Met (+12.88%), and Ile (+5.98%) all increase. Conversely, His (–5.26%) and Trp (–8.67%) decrease.

The dynamics of CP and essential AA deposition in the pregnant gilt predicted by our model are novel in that previous models did not account for the dynamic changes in tissue composition throughout gestation. The NRC (2012) [1] gestating sow model utilized average CP and Lys concentrations during the entire gestation period for each tissue, without considering these temporal variations.

## 4. Discussion

### 4.1. Early Gestation

The consistent observation of early gestation weight loss across several studies (Buis [46], Thomas et al. [47], Stewart [48], and Ji et al. [2]) suggests that tissue mobilization during the first weeks of gestation may be a common gilt response. Although the precise mechanisms remain unclear, one plausible explanation is that increased AA demands for uterine fluid production drive this response. Uterine fluids, which are the primary source of nutrition for porcine embryos [50], require high levels of certain essential AAs despite comprising only a small fraction of total body tissue. For example, metabolites such as taurine, biosynthesized from Cys and Met [51], are present in uterine fluids at concentrations 190-fold higher than Met and 65 times greater than Lys on day 5 after estrus [52]. Therefore, the elevated demand for sulfur-containing AAs for taurine biosynthesis (or potentially other essential AAs) may contribute to the observed weight loss during early gestation. Further research is needed to rigorously test this hypothesis.

Nevertheless, regardless of the underlying cause, Figure 7A shows that the deficit in daily protein deposition during the early days of gestation was calculated to be approximately 300 g CP/day. If we consider common corn-soybean meal-based gestation diets that provide between 15% and 12% digestible CP, the additional 300 g CP/d would be equivalent to providing an additional 2 to 2.5 kg of feed/d. Under current commercial practices, this would correspond to providing twice the current feeding levels during the first week of gestation.

The notion that increased feeding levels are needed during early gestation has been established by previous models such as the Brazilian Table of Nutrient Requirements published by Rostagno et al. [53]. In addition, increased needs for CP intake are substantiated by various authors who have demonstrated that increased AA intake during the initial weeks of gestation enhances luteal tissue mass, embryo survival, and the sustenance of pregnancy, as reviewed by Langendijk [54].

The current study suggests that body weight loss occurs in pregnant gilts during early gestation. Similarly, weight loss appears to occur in older parities, as evidenced by data reported by Stewart [48], which shows weight loss during the initial two weeks of gestation in multiparous sows (Figure 12). While the weight loss in multiparous sows could be partly linked to mammary gland regression after lactation rather than a dietary deficiency, this factor only provides a partial explanation. According to Ford Jr et al. [55], the mammary gland reduces approximately 70% of its weight during the first 7 days post-weaning. Specifically, Ford Jr et al. [55] reported that during the first week after weaning, the mammary gland decreased in weight from 486 to 152 g per gland, implying a reduction from 6.8 to 2.1 kg if considering a sow with 14 functional teats. Because sows are typically bred around day 7 post-weaning, the remaining ~2.1 kg of mammary tissue is expected to regress further during gestation. Nonetheless, the average weight loss for parities 2 and 3 sows reported by Stewart [48] was 8 kg. Thus, the regression of the mammary gland after lactation could explain approximately 25% of the reduction in body weight during early gestation in multiparous sows. The remaining weight loss may be associated with the mobilization of other tissues, potentially driven by deficiencies in AA, protein, and/or other nutrients.

The previous discussion suggests that during early gestation, pregnant gilts and sows may require increased dietary AA, a finding that contrasts with recommendations from models such as the NRC (2012) [1] gestating sow model. Our calculations indicate that previous recommendations may be influenced by model-dependent biases, where the inherent assumptions and structure of the model drive the outcomes more than direct empirical observations. In earlier models, assumptions regarding CP deposition and nutrient partitioning during early gestation may have led to recommendations that do not fully reflect the biological response. By revisiting these assumptions and incorporating updated datasets, our model suggests that increased dietary AA during early gestation may indeed be necessary. Addressing this potential bias in data modeling helps clarify how our model contributes to a more accurate understanding of early gestation growth and CP and AA deposition.

#### Time-Dependent Protein Deposition

An important difference between our proposed model and the NRC (2012) [1] gestating sow model is the protein pool termed “time-dependent protein deposition,” which our model does not include. According to the NRC (2012) [1], time-dependent protein deposition refers to an increase in CP deposition in the maternal body during early gestation, peaking around day 40 (Figure 10B). However, our findings suggest that, rather than peaking at day 40, CP deposition is underestimated during the first three weeks of gestation. This underestimation appears to be related to the assumptions made and the choice of model employed during mathematical modeling and/or in interpreting nitrogen balance data.

As reported by the NRC (2012) [1], their CP deposition model in pregnant sows was developed based on a comprehensive literature review that spanned from the years 1965 to 2003. Nevertheless, no data were available for the first weeks of gestation, and assumptions had to be made. The NRC (2012) [1] assumed reduced CP deposition during the first days of gestation. As shown in Figure 13A, it seems that this assumption of reduced CP deposition during the first days of gestation caused the emergence of time-dependent protein deposition. Considering this reduced early CP deposition assumption and additional refinements, the NRC (2012) [1] CP deposition model was developed (Figure 13B). Nevertheless, using the same data on which the NRC (2012) [1] CP deposition model was developed, but when the assumption of reduced CP deposition around breeding is not considered, the relationship between the day of gestation and CP deposition becomes quadratic, and the time-dependent protein deposition pool disappears (Figure 13C).

Although the current study utilized data from a single experiment to develop the whole-body protein deposition curve across gestation, the compiled data from all reviewed studies further support the notion of increased protein deposition during early gestation. The combined whole-body protein deposition data in Figure 14 demonstrate a statistically significant quadratic regression (*p* = 0.001), indicating elevated protein deposition during early gestation. Conversely, a cubic regression, which would suggest a reduction in protein deposition during early gestation, was not statistically significant (*p* = 0.199). Thus, incorporating a cubic component does not enhance the model’s fit, confirming the superiority of the quadratic model. These findings suggest that protein deposition is greater in early gestation than in mid-gestation, posing a challenge to existing models like the NRC (2012) [1] gestating sow model.

Nevertheless, the NRC (2012) [1] assumption of reduced CP deposition during the first weeks of gestation does not appear to be arbitrary, as it is supported by empirical data. As described in Figure 8 and Figure 12, data from multiple studies show a negative weight gain during the first weeks of gestation. This body weight loss would result in negative CP retention, which, in turn, would increase CP excretion as endogenous protein is catabolized. Since dietary CP retention is calculated as CP intake minus CP excretion, this potential increase in endogenous CP excretion can be interpreted as reduced CP deposition.

Our interpretation is not that the CP deposition potential of the animal is reduced during early gestation, but rather that the animal may be mobilizing maternal protein to obtain AAs that are necessary to meet reproductive needs, potentially due to a deficiency in specific essential AAs or CP intake. If current feeding practices do not supply adequate essential AAs or protein during the initial weeks of gestation, the gestating gilt might make use of body tissues leading to weight loss and the excretion of excess AAs that cannot be utilized for protein synthesis. Thus, reduced CP deposition during early gestation may reflect nutrient deficiency rather than diminished protein synthesis capacity. We speculate that increasing levels of specific AAs during early gestation could minimize maternal tissue mobilization and potentially promote positive protein deposition.

The observed body weight loss during early gestation occurred in animals consuming the same diet throughout gestation, suggesting that a diet relatively adequate for mid- and late gestation may not be sufficient for early gestation. However, it is not yet clear which specific AAs or whether increased dietary CP is required during early gestation. As reviewed by Langendijk [54], “changes in the concentrations of Leu, Arg, glutamine, glucose, and fructose in uterine and trophoblast fluids during blastocyst development and trophoblast elongation suggest these molecules may act as functional nutrients for early embryo development.” Therefore, pregnant gilts may require increased levels of specific AAs and/or protein, along with additional energy during early gestation, suggesting that the AA profile of dietary protein in the first weeks of gestation may need to differ from that required for the remainder of the gestation period.

### 4.2. Mid-Gestation

The current model was developed using daily whole-body protein deposition data from Miller et al. [9]. In the absence of direct data for early gestation, we adopted the assumption of reduced protein deposition suggested by previous models (see Figure 2A). However, if this assumption is removed, the pattern of daily whole-body protein deposition becomes quadratic (Figure 15).

By updating our model under a quadratic whole-body protein deposition trend, the essential AA deposition model also becomes quadratic (Figure 16). This modeling exercise suggests that addressing the potential nutritional deficiency (or other means to minimize weight loss) during early gestation could enable the pregnant gilt to maintain a relatively constant daily level of Lys throughout gestation. However, the model-predicted essential AA deposition in Figure 16 and empirical data (Figure 13C, Figure 14 and Figure 15) still show a reduction in AA and CP deposition during mid-gestation.

Our working hypothesis is that the potential reduction in CP deposition during mid-gestation can be attributed to two different phenomena: an increase in energy storage and an increase in stem cell storage. As reported by Samuel [60], during mid-gestation pregnant sows seem to have a primary need for energy storage rather than AA. During mid-gestation, placental glycogen stores increase, which is assumed to ensure that the fetal glucose supply is maintained at times of maximum demand such as during late gestation [61,62,63]. In addition to the explanation on energy, there is the potential role of stem cell development; experiments in multiple species have revealed that amniotic fluid contains stem cells [64,65,66], and experiments in humans and mice have revealed that the proportion of amniotic stem cells peaks during mid-gestation and decreases thereafter [65,67,68]. Because amniotic stem cells can differentiate into somatic cells (i.e., bone, fat, cartilage, muscle, hematopoietic, endothelial, hepatic, and neuronal tissues) [65], it is speculated that the peak production of amniotic stem cells during mid-gestation occurs to support the rapid fetal growth that occurs during late gestation [69]. Thus, the body may reduce CP deposition during mid-gestation to prioritize placental fat deposition and amniotic fluid stem cell production.

### 4.3. Late Gestation and the Entire Gestation Period

Our results show that the AA profile of the protein deposited during late gestation varies in relation to mid-gestation and potentially to early gestation. Specifically, AA ratios relative to Lys increase for all essential AAs except His and Trp. These observations suggest that ideal dietary protein requirements during late gestation may require incremental adjustments in the ratios of most essential AAs relative to Lys. However, the exact ratios need to be empirically established. Nonetheless, our findings provide a foundation for future research on the specific AA requirements during early gestation.

Overall, our results suggest that a single diet is unlikely to meet all the needs of gilts throughout the entire gestation period, as physiological needs evolve. During early gestation, the body may prioritize uterine fluid production (or other unknown functions) over protein synthesis, as indicated by the observed weight loss. In mid-gestation, the body may prioritize placenta development and stem cell production over maternal or fetal protein synthesis, as indicated by the reduction in whole-body CP retention that coincides with the peak growth of the placenta and fluids (Figure 1A,D,G). Additionally, during late gestation, the profile of synthesized whole-body protein shows differences compared to early and mid-gestation.

Our findings suggest that models of CP and essential AA deposition may be insufficient for establishing dietary AA requirements during gestation if they do not account for the needs of tissues such as uterine and amniotic fluids. Although these fluids do not represent a significant mass of CP and AAs on a whole-body weight basis, their turnover rates must be considered. The biosynthesis of metabolites found in these fluids may require a high daily amount of specific essential AAs, which must be provided by the diet to ensure optimal animal performance. It is important to recognize that AAs not only serve a structural role in the body but are also essential for various metabolic functions.

## 5. Conclusions

Our models describe daily CP and essential AA deposition in pregnant gilts throughout gestation. By incorporating previously unconsidered factors, such as dynamic changes in tissue CP and essential AA composition, our approach reveals potential mismatches between AA intake, retention, and tissue growth rates with consequent nutritional requirements of gilts at different gestational stages. Our comparative analysis with the NRC (2012) [1] gestating sow model highlights how our findings fit within the current literature while identifying critical knowledge gaps. This comparison highlights the limitations of existing models and the need for further research to refine dietary recommendations that better accommodate the evolving metabolic demands of gestation. Specifically, our results suggest that dietary AA profiles and CP levels should be adjusted to meet the metabolic demands of early and mid-gestation and the increased essential AA demands during late gestation. Overall, our models provide a more detailed perspective on protein and AA deposition throughout gestation. These predictions can serve as a foundation for in vivo studies aimed at optimizing nutritional strategies that enhance both the performance and well-being of swine during pregnancy.

## Figures and Tables

**Figure 1 animals-15-02126-f001:**
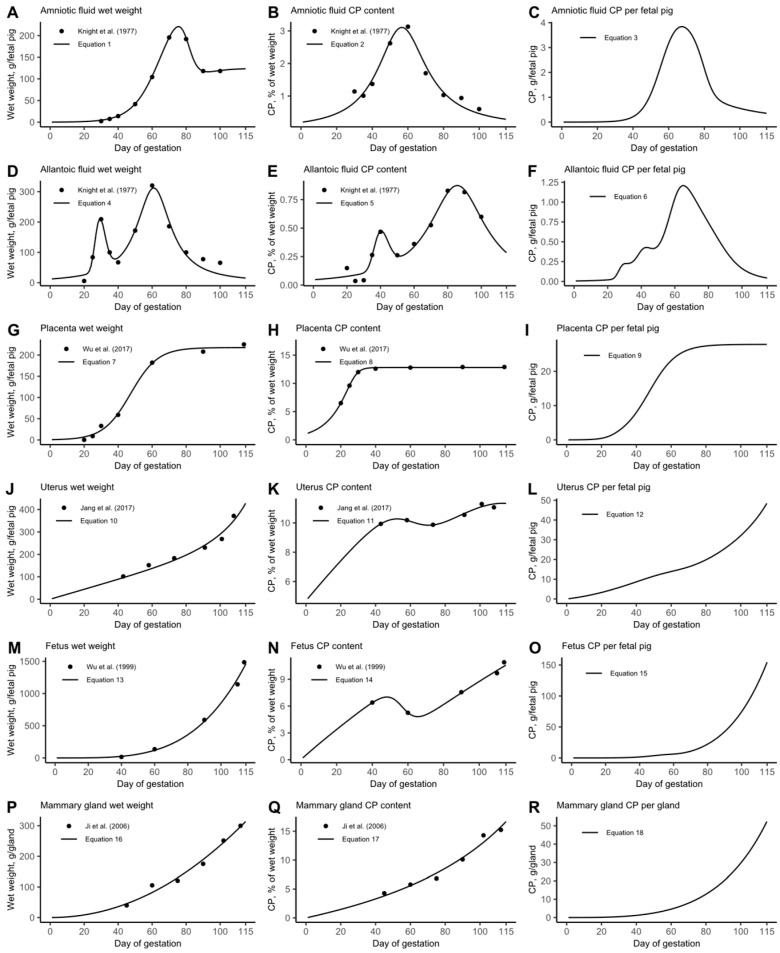
Graphic representation of wet weight growth curves and crude protein (CP) composition curves developed for key tissues in the model. Data extracted from Knight et al. (1977) [12], Wu et al. (2017) [14], Jang et al. (2017) [3], Wu et al. (1999) [4], and Ji et al. (2006) [24].

**Figure 2 animals-15-02126-f002:**
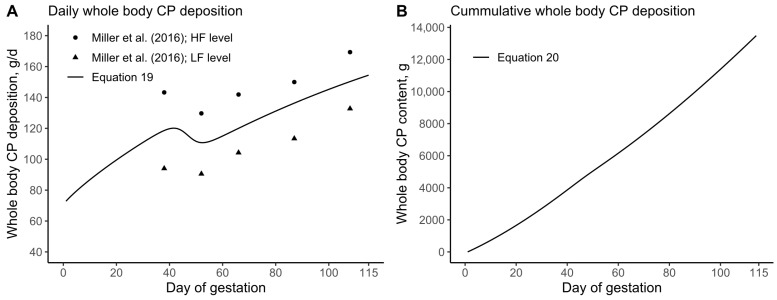
(**A**) Daily whole-body crude protein (CP) deposition; data extracted from Miller et al. (2016) [9]; (**B**) cumulative sum of the daily whole-body CP deposition in the pregnant gilt. HF = High feeding, LF = Low feeding.

**Figure 3 animals-15-02126-f003:**
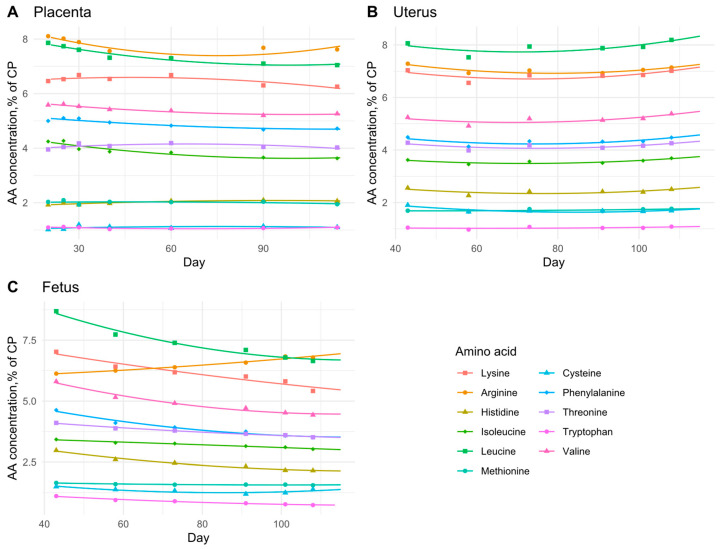
Essential amino acid (AA) concentrations as a proportion of crude protein (CP) across gestation in (**A**) placenta, data extracted from Wu et al. [14], (**B**) uterus, data extracted from Jang et al. [3], and (**C**) fetus, data extracted from Jang et al. [3].

**Figure 4 animals-15-02126-f004:**
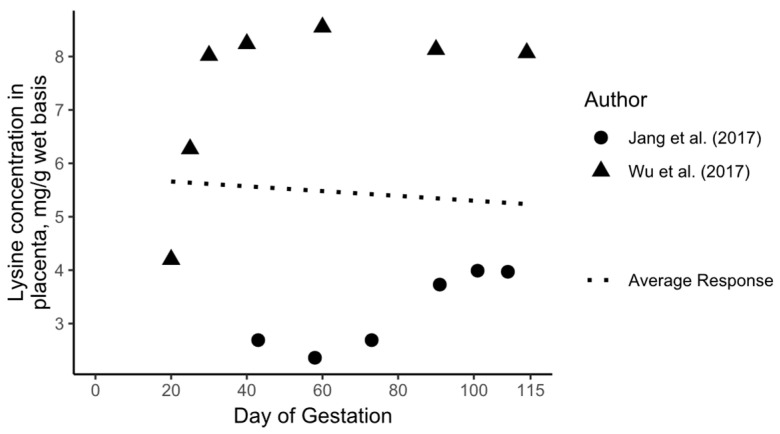
Lysine concentrations in the placenta throughout gestation. Data extracted from Jang et al. (2017) [3] and Wu et al. (2017) [14].

**Figure 5 animals-15-02126-f005:**
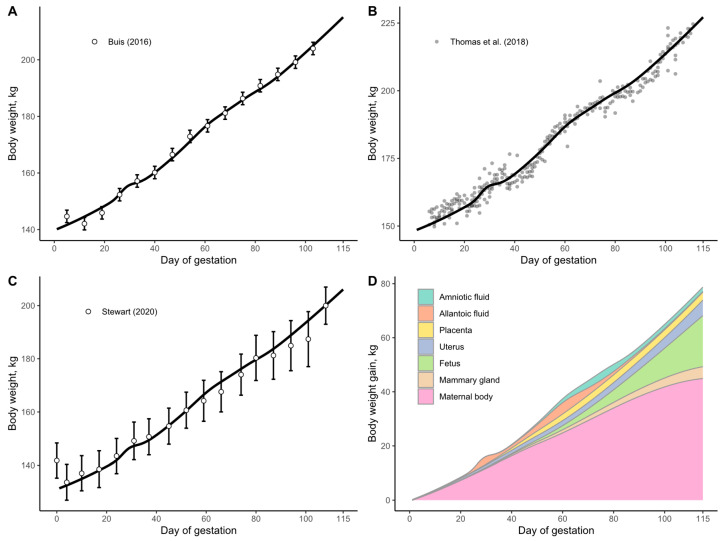
Model-predicted body weight gain (solid lines) compared to observed body weights reported by (**A**) Buis [46]; model parameterized as litter size = 12.4, average birth weight = 1.5 kg based on the observed farrowing performance. (**B**) Thomas et al. [47]; model parameterized as litter size = 14.9, average birth weight = 1.2 kg based on the farrowing performance reported by Thomas [49]. (**C**) Stewart [48]; model parameterized as litter size = 12, average birth weight = 1.35 kg based on the observed farrowing performance. For all three datasets, 14 available teats were assumed. Reported initial body weights were not utilized; instead, the initial body weight inputs were the values that minimize the mean square error of the model’s predicted values. (**D**) Body weight gain partitioned by tissue; calculated for the Buis [46] dataset. Error bars represent the standard error. The extracted data from Buis [46] and Stewart [48] correspond to animals in the control group during the first parity of the experiment.

**Figure 6 animals-15-02126-f006:**
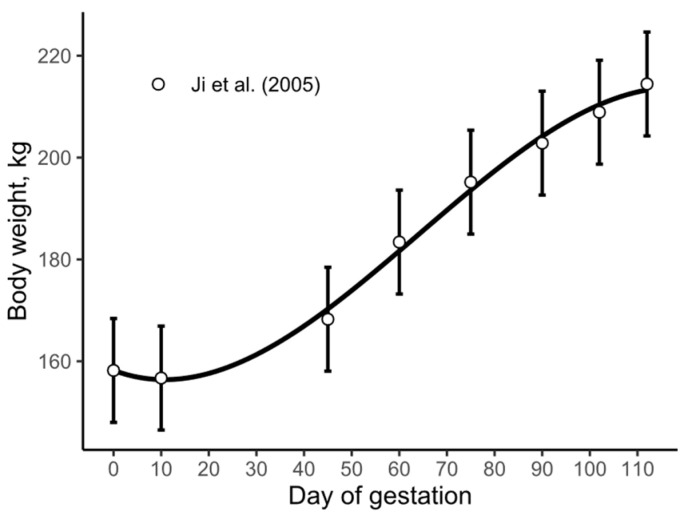
Body weights of gilts allocated in stalls during gestation; data extracted from Ji et al. [2]. Solid line represents a cubic model; the developed gestating sow model was not used for this data as animals of each group had different reproductive performance. Error bars represent the standard error.

**Figure 7 animals-15-02126-f007:**
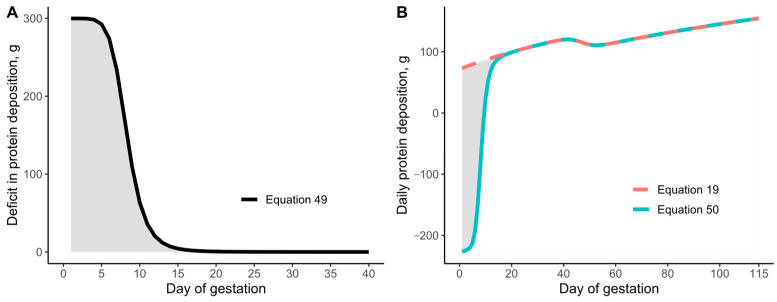
Adjustment of the daily protein deposition curve to accommodate the observed negative (deficit) protein deposition in gilts during early gestation. (**A**) Calculated negative protein deposition during early gestation (deficit). (**B**) Recalculated daily protein deposition Equation (50) considering the negative protein deposition observed during early gestation.

**Figure 8 animals-15-02126-f008:**
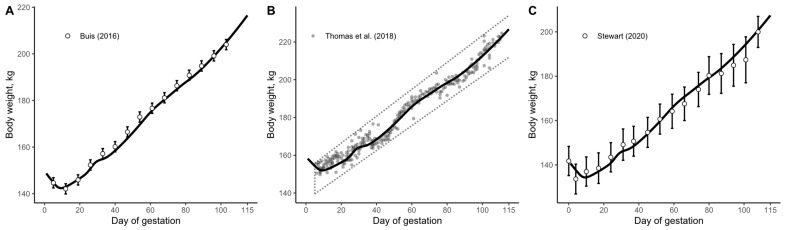
The updated model-predicted body weight gain (solid lines) accounting for early gestation body weight loss, compared to observed body weights reported by (**A**) Buis [46], (**B**) Thomas et al. [47], and (**C**) Stewart [48]. Body weights lying outside the dotted line area in panel B were considered inaccurate by the authors.

**Figure 9 animals-15-02126-f009:**
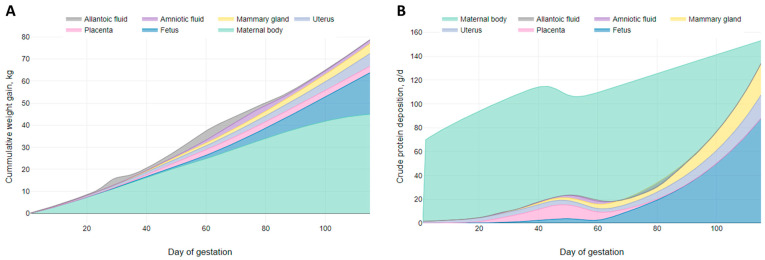
Visual representation of (**A**) the growth model by tissue; (**B**) the crude protein deposition model by tissue. Models parameterized with litter size = 14.9, average piglet birth weight = 1.2 kg, available teats = 14, and initial body weight = 150 kg.

**Figure 10 animals-15-02126-f010:**
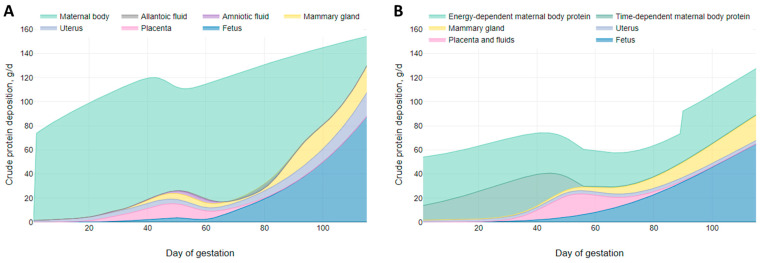
Visual representation of (**A**) our proposed crude protein deposition model by tissue (parameterized with litter size = 13.5, average piglet birth weight = 1.4 kg, available teats = 14, and initial body weight = 150 kg); (**B**) the NRC (2012) [1] crude protein deposition model [1] by tissue (parameterized with litter size = 13.5, average piglet birth weight = 1.4 kg, parity = 1, and initial body weight = 150 kg).

**Figure 11 animals-15-02126-f011:**
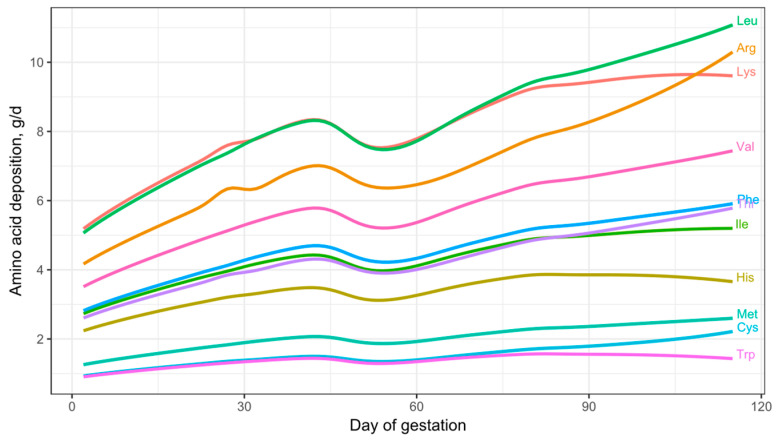
Visual representation of the daily essential amino acid deposition model for gilts across gestation. The model was parameterized with litter size = 13.5, average piglet birth weight = 1.4 kg, available teats = 14, and initial body weight = 150 kg.

**Figure 12 animals-15-02126-f012:**
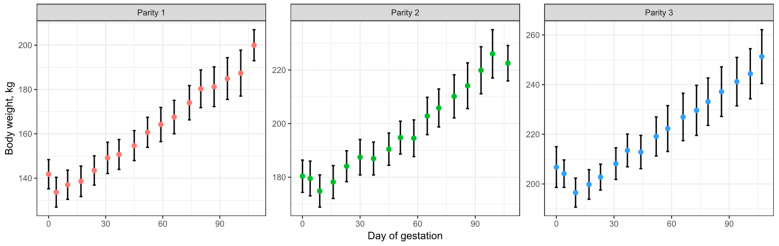
Body weight loss in multiparous sows occurs during early gestation; data extracted from Stewart [48]. The extracted data refers to animals in the control group during the first parity cycle of the experiment.

**Figure 13 animals-15-02126-f013:**
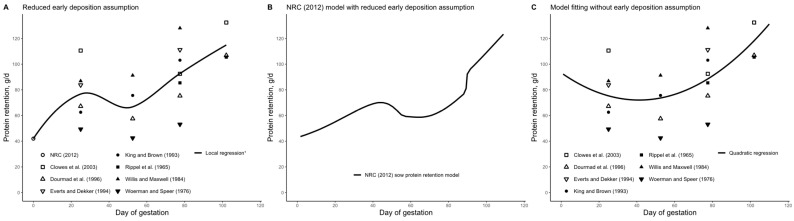
(**A**) The NRC (2012) [1] developed a model of crude protein deposition during pregnancy based on a literature review, but it assumed low protein retention during the pre-implantation period. (**B**) The protein retention model was refined relative to the raw data which resulted in the final NRC (2012) crude protein deposition model [1]. (**C**) In the absence of the assumption of low protein retention during the pre-implantation period, the protein retention pattern becomes quadratic. The NRC (2012) [1] crude protein retention model was developed using data from Clowes et al. (2003) [26], Dourmad et al. (1996) [56], Everts and Dekker (1994) [57], King and Brown (1993) [30], Rippel et al. (1965) [58], Willis and Maxwell (1984) [35], and Woerman and Speer [59]. ^1^ Local regression is a method for fitting a smooth curve through points and is used to reveal patterns in the data.

**Figure 14 animals-15-02126-f014:**
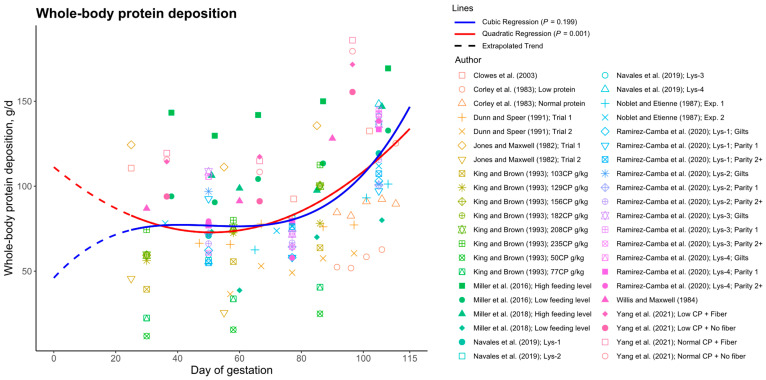
Whole-body protein deposition data across gestation in sows, compiled from a systematized search of studies reporting protein deposition at more than three time points during gestation. A quadratic regression model demonstrated a statistically significant fit to the data (*p* = 0.001), whereas a cubic regression model failed to reach statistical significance (*p* = 0.199). Data extracted from Clowes et al. (2003) [26], Corley et al. (1983) [27], Dunn and Speer (1991) [28], Jones and Maxwell (1982) [29], King and Brown (1993) [30], Miller et al. (2016) [9], Miller et al. (2018) [31], Navales et al. (2019) [32], Noblet and Etienne (1987) [33], Ramirez-Camba et al. (2020) [34], Willis and Maxwell (1984) [35], and Yang et al. (2021) [36]. One protein deposition value reported in Jones and Maxwell (1982) [29] was negative and was excluded from the analysis.

**Figure 15 animals-15-02126-f015:**
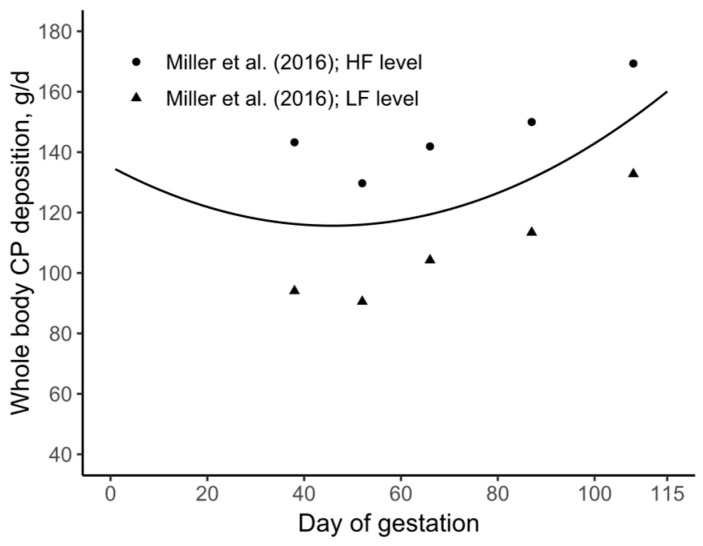
Daily whole-body CP deposition modeled without the assumption of reduced CP deposition during early gestation. Data extracted from Miller et al. [9]. HF = High feeding, LF = Low feeding.

**Figure 16 animals-15-02126-f016:**
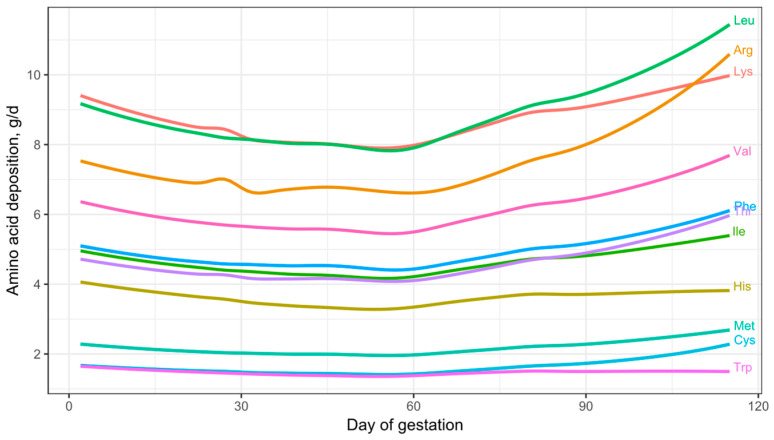
Visual representation of the daily essential amino acid deposition model for gilts across gestation, using the recalculated model without the assumption of reduced CP deposition during early gestation. The model was parameterized with litter size = 13.5, average piglet birth weight = 1.4 kg, available teats = 14, and initial body weight = 150 kg.

**Table 4 animals-15-02126-t004:** Equations used in the cumulative growth model.

$\mathrm{Amniotic}\;\mathrm{fluid}\;(\mathrm{WW}),\;g\cdot {\mathrm{fetal}\;\mathrm{pig}}^{-1}=\frac{321.22}{\;1+2400\cdot \exp\left(-0.118\cdot \mathrm{day}\right)}-\frac{196.86}{1+{\left(\frac{\mathrm{day}}{81.4}\right)}^{-27.83}}$	(1)
$\mathrm{Amniotic}\;\mathrm{fluid}\;\mathrm{CP}\;\mathrm{content},\;\%\;*=\frac{0.1928+0.00107\cdot \mathrm{day}}{1\;-\;0.0328\cdot \mathrm{day}+2.928\;\times {\;10}^{-4}{\cdot \mathrm{day}}^{2}}$	(2)
$\mathrm{Amniotic}\;\mathrm{fluid}\;\mathrm{CP},\;g\cdot {\mathrm{fetal}\;\mathrm{pig}}^{-1}=\;\mathrm{Equation}\;(1)\;\times \;\mathrm{Equation}\;(2)/100$	(3)
$\mathrm{Allantoic}\;\mathrm{fluid}\;(\mathrm{WW}),\;g\cdot {\mathrm{fetal}\;\mathrm{pig}}^{-1\;}=\exp\left(420.40\;-\frac{2774.5}{\mathrm{day}}\;-\;94.91\cdot \ln\left(\mathrm{day}\right)\right)+\frac{1}{0.0823+\left(-2.596\;\times \;10^{-3}\right)\cdot \mathrm{day}+2.13\;\times \;10^{-5}{\cdot \mathrm{day}}^{2}}$	(4)
$\mathrm{Allantoic}\;\mathrm{fluid}\;\mathrm{CP}\;\mathrm{content},\;\%\;*=\;\exp\left(419.15\;-\frac{3595.38}{\mathrm{day}}-\;89.56\cdot \mathrm{day}\cdot \ln\left(\mathrm{day}\right)\right)+\frac{1}{21.45\;-\;0.4739\cdot \mathrm{day}+0.002765{\cdot \mathrm{day}}^{2}}$	(5)
$\mathrm{Allantoic}\;\mathrm{fluid}\;\mathrm{CP},\;g\cdot {\mathrm{fetal}\;\mathrm{pig}}^{-1}=\;\mathrm{Equation}\;(4)\;\times \;\mathrm{Equation}\;(5)/100$	(6)
$\mathrm{Placenta}\;(\mathrm{WW}),\;g\cdot {\mathrm{fetal}\;\mathrm{pig}}^{-1}=\frac{217.504}{1+\exp\;(5.896\;-\;0.1247\cdot \mathrm{day})}$	(7)
$\mathrm{Placenta}\;\mathrm{CP}\;\mathrm{content},\;\%\;*=\;\frac{12.808}{{(1+\exp\;(10.943\;-\;0.4005\cdot \mathrm{day})}^{0.2274}}$	(8)
$\mathrm{Placenta}\;\mathrm{CP},\;g\cdot {\mathrm{fetal}\;\mathrm{pig}}^{-1}=\;\mathrm{Equation}\;(7)\;\times \;\mathrm{Equation}\;(8)/100$	(9)
$\mathrm{Uterus}\;(\mathrm{WW}),\;g\cdot {\mathrm{fetal}\;\mathrm{pig}}^{-1}=\frac{\mathrm{day}}{0.451\;-\;1.98\;\times \;10^{-9}\cdot {\mathrm{day}}^{3.865}}$	(10)
$\mathrm{Uterus}\;\mathrm{CP}\;\mathrm{content},\;\%\;*=5.277+0.147\cdot \mathrm{day}-\frac{4.606}{1+2093\cdot \exp\;(-0.127\cdot \mathrm{day})}-\;0.598{\cdot \mathrm{day}}^{0.0043\cdot \mathrm{day}}$	(11)
$\mathrm{Uterus}\;\mathrm{CP},\;g\cdot {\mathrm{fetal}\;\mathrm{pig}}^{-1\;}=\;\mathrm{Equation}\;(10)\;\times \;\mathrm{Equation}\;(11)/100$	(12)
$\mathrm{Fetus}\;(\mathrm{WW}),\;g\cdot {\mathrm{fetal}\;\mathrm{pig}}^{-1}=1.63\;\times \;10^{5}\cdot {\mathrm{day}}^{3.859}$	(13)
$\mathrm{Fetus}\;\mathrm{CP}\;\mathrm{content},\;\%\;*=9.286+0.194\cdot \mathrm{day}-\frac{6.027}{1+817735\cdot \exp\;(-0.24\cdot \mathrm{day})}-\;9.277{\cdot \mathrm{day}}^{8.8\times 10^{-4}\cdot \mathrm{day}}$	(14)
$\mathrm{Fetus}\;\mathrm{CP},\;g\cdot {\mathrm{fetal}\;\mathrm{pig}}^{-1}\cdot {\mathrm{day}}^{-1\;}=\;\mathrm{Equation}\;(13)\;\times \;\mathrm{Equation}\;(14)/100$	(15)
$\mathrm{Mammary}\;\mathrm{gland}\;(\mathrm{WW}),\;g\cdot {\mathrm{gland}}^{-1}=0.0178\;\times \;{\mathrm{day}}^{2.0607}$	(16)
$\mathrm{Mammary}\;\mathrm{gland}\;\mathrm{CP},\;\%\;*=-14.324\;\times \;\frac{\mathrm{day}}{-213.737+\mathrm{day}}$	(17)
$\mathrm{Mammary}\;\mathrm{gland}\;\mathrm{CP},\;g\cdot {\mathrm{gland}}^{-1}=\;\mathrm{Equation}\;(16)\;\times \;\mathrm{Equation}\;(17)/100$	(18)
$\mathrm{Daily}\;\mathrm{whole-body}\;\mathrm{CP}\;\mathrm{retention},\;g=\;841.7+2.14\cdot \mathrm{day}-\frac{25.24}{1+8.67\;\times \;10^{7}\cdot \exp\left(-0.39\cdot \mathrm{day}\right)}-771{\cdot \mathrm{day}}^{0.0003\cdot \mathrm{day}}$	(19)
$\mathrm{Cummulative}\;\mathrm{whole-body}\;\mathrm{CP}\;\mathrm{retention},\;g\;=\;\int_{1}^{\mathrm{day}}\mathrm{Equation}\;(19)$	(20)
$\mathrm{Essential}\;\mathrm{AA}\;\mathrm{deposition},\;g/100\;\mathrm{CP}=a+b\;\cdot \;\mathrm{day}+c\;\cdot \;{day}^{2}$	(21)
$\mathrm{Conceptus}\;\mathrm{and}\;\mathrm{Mammary}\;\mathrm{gland}\;\mathrm{CP},\;g=\mathrm{Litter}\;\mathrm{size}\cdot \left(\mathrm{Equation}\;(3)\;+\;\mathrm{Equation}\;(6)\;+\;\mathrm{Equation}\;(9)\;+\;\mathrm{Equation}\;(12)\;+\;\mathrm{Equation}\;(15)\cdot \frac{\mathrm{Piglet}\;\mathrm{birth}\;\mathrm{weight}}{1.46023}\right)+\mathrm{Available}\;\mathrm{teats}\cdot \mathrm{Equation}\;(18)$	(22)
$\mathrm{Maternal}\;\mathrm{body}\;(\mathrm{WW}),\;\mathrm{kg}=\frac{\frac{\mathrm{Equation}\;(20)\;-\;\mathrm{Equation}\;(22)}{1000}}{0.22}$	(23)
$\mathrm{Whole}\;\mathrm{body}\;\left(\mathrm{WW}\right),\;\mathrm{kg}=\frac{\mathrm{Litter}\;\mathrm{size}\cdot \left(\mathrm{Equation}\;(1)\;+\;\mathrm{Equation}\;(4)\;+\;\mathrm{Equation}\;(7)\;+\;\mathrm{Equation}\;(10)\;+\;\mathrm{Equation}\;(13)\cdot \frac{\mathrm{Piglet}\;\mathrm{birth}\;\mathrm{weight}}{1.46023}\right)+\mathrm{Available}\;\mathrm{teats}\cdot \mathrm{Equation}\;(16)}{1000}+\;\mathrm{Equation}\;(23)$ + Initialweight	(24)

Input variables used in these equations: day: day of gestation; litter size: number of pigs born; piglet birth weight: average piglet birth weight in kg; available teats: number of available teats per gilt; initial weight: body weight at breeding in kg. * Percentage of tissue wet weight. WW = wet weight basis.

**Table 5 animals-15-02126-t005:** Coefficients for constructing quadratic functions representing essential amino acid (AA) concentrations in the placenta, uterus, and fetus.

Amino Acid	Placenta			Uterus			Fetus		
	a	b	c	a	b	c	a	b	c
Lysine	6.392 × 10^0^	8.575 × 10^−3^	−9.008 × 10^−5^	8.130 × 10^0^	−3.873 × 10^−2^	2.632 × 10^−4^	8.165 × 10^0^	−3.176 × 10^−2^	7.172 × 10^−5^
Arginine	8.575 × 10^0^	−3.387 × 10^−2^	2.242 × 10^−4^	8.519 × 10^0^	−4.057 × 10^−2^	2.575 × 10^−4^	5.900 × 10^0^	2.906 × 10^−3^	5.373 × 10^−5^
Histidine	1.824 × 10^0^	5.548 × 10^−3^	−2.984 × 10^−5^	3.214 × 10^0^	−2.299 × 10^−2^	1.518 × 10^−4^	4.060 × 10^0^	−3.131 × 10^−2^	1.273 × 10^−4^
Isoleucine	4.561 × 10^0^	−1.869 × 10^−2^	9.358 × 10^−5^	4.213 × 10^0^	−2.015 × 10^−2^	1.402 × 10^−4^	3.634 × 10^0^	−5.129 × 10^−3^	−2.704 × 10^−6^
Leucine	8.243 × 10^0^	−2.432 × 10^−2^	1.237 × 10^−4^	9.264 × 10^0^	−4.330 × 10^−2^	3.061 × 10^−4^	1.138 × 10^1^	−7.915 × 10^−2^	3.332 × 10^−4^
Methionine	1.996 × 10^0^	1.483 × 10^−3^	−1.556 × 10^−5^	1.721 × 10^0^	−1.554 × 10^−3^	1.718 × 10^−5^	1.801 × 10^0^	−4.641 × 10^−3^	2.314 × 10^−5^
Cysteine	1.003 × 10^0^	3.290 × 10^−3^	−2.163 × 10^−5^	2.598 × 10^0^	−2.283 × 10^−2^	1.352 × 10^−4^	2.331 × 10^0^	−2.528 × 10^−2^	1.480 × 10^−4^
Phenylalanine	5.282 × 10^0^	−1.015 × 10^−2^	4.402 × 10^−5^	5.359 × 10^0^	−3.044 × 10^−2^	2.054 × 10^−4^	6.099 × 10^0^	−4.309 × 10^−2^	1.799 × 10^−4^
Threonine	3.870 × 10^0^	8.746 × 10^−3^	−6.729 × 10^−5^	4.997 × 10^0^	−2.492 × 10^−2^	1.666 × 10^−4^	4.677 × 10^0^	−1.591 × 10^−2^	4.996 × 10^−5^
Tryptophan	1.162 × 10^0^	−3.337 × 10^−3^	2.441 × 10^−5^	1.119 × 10^0^	−3.121 × 10^−3^	2.485 × 10^−5^	1.543 × 10^0^	−1.244 × 10^−2^	4.742 × 10^−5^
Valine	5.832 × 10^0^	−1.189 × 10^−2^	5.929 × 10^−5^	6.013 × 10^0^	−2.781 × 10^−2^	2.009 × 10^−4^	7.709 × 10^0^	−5.614 × 10^−2^	2.431 × 10^−4^

a, b, and c represent the coefficients of the quadratic regression: AA concentration, % of CP = a + b·x + c·x^2^.

**Table 6 animals-15-02126-t006:** Equations used in the daily crude protein (CP) deposition model.

$\mathrm{Daily}\;\mathrm{Amniotic}\;\mathrm{fluid}\;\mathrm{CP}\;\mathrm{deposition},\;g\cdot {\mathrm{fetal}\;\mathrm{pig}}^{-1}\cdot {\mathrm{day}}^{-1}=\mathrm{Equation}\;(3)\;(\mathrm{day}+1)-\mathrm{Equation}\;(3)\;(\mathrm{day});\;\mathrm{for}\;\mathrm{day}=1\;\mathrm{to}\;114$	(25)
$\mathrm{Daily}\;\mathrm{Allantoic}\;\mathrm{fluid}\;\mathrm{CP}\;\mathrm{deposition},\;g\cdot {\mathrm{fetal}\;\mathrm{pig}}^{-1}\cdot {\mathrm{day}}^{-1}=\mathrm{Equation}\;(6)\;(\mathrm{day}+1)-\mathrm{Equation}\;(6)\;(\mathrm{day});\;\mathrm{for}\;\mathrm{day}=1\;\mathrm{to}\;114$	(26)
$\mathrm{Daily}\;\mathrm{Placenta}\;\mathrm{CP}\;\mathrm{deposition},\;g\cdot {\mathrm{fetal}\;\mathrm{pig}}^{-1}\cdot {\mathrm{day}}^{-1}=\mathrm{Equation}\;(9)\;(\mathrm{day}+1)-\mathrm{Equation}\;(9)\;(\mathrm{day});\;\mathrm{for}\;\mathrm{day}=1\;\mathrm{to}\;114$	(27)
$\mathrm{Daily}\;\mathrm{Uterus}\;\mathrm{CP}\;\mathrm{deposition},\;g\cdot {\mathrm{fetal}\;\mathrm{pig}}^{-1}\cdot {\mathrm{day}}^{-1}=\mathrm{Equation}\;(12)\;(\mathrm{day}+1)-\mathrm{Equation}\;(12)\;(\mathrm{day});\;\mathrm{for}\;\mathrm{day}=1\;\mathrm{to}\;114$	(28)
$\mathrm{Daily}\;\mathrm{Fetus}\;\mathrm{CP}\;\mathrm{deposition},\;g\cdot {\mathrm{fetal}\;\mathrm{pig}}^{-1}\cdot {\mathrm{day}}^{-1}=\mathrm{Equation}\;(15)\;(\mathrm{day}+1)-\mathrm{Equation}\;(15)\;(\mathrm{day});\;\mathrm{for}\;\mathrm{day}=1\;\mathrm{to}\;114$	(29)
$\mathrm{Daily}\;\mathrm{Mammary}\;\mathrm{gland}\;\mathrm{CP}\;\mathrm{deposition},\;g\cdot {\mathrm{gland}}^{-1}\cdot {\mathrm{day}}^{-1\;}=\mathrm{Equation}\;(18)\;(\mathrm{day}+1)-\mathrm{Equation}\;(18)\;(\mathrm{day});\;\mathrm{for}\;\mathrm{day}=1\;\mathrm{to}\;114$	(30)
Maternal body CP, g = Equation (23) ·1000· 0.22	(31)
$\mathrm{Daily}\;\mathrm{Maternal}\;\mathrm{body}\;\mathrm{CP}\;\mathrm{deposition},\;g\cdot {\mathrm{day}}^{-1}=\mathrm{Equation}\;(31)\;(\mathrm{day}+1)-\mathrm{Equation}\;(31)\;(\mathrm{day});\;\mathrm{for}\;\mathrm{day}=1\;\mathrm{to}\;114$	(32)
$\mathrm{Daily}\;\mathrm{Amniotic}\;\mathrm{fluid}\;\mathrm{CP}\;\mathrm{deposition},\;g\cdot {\mathrm{day}}^{-1}=\mathrm{Equation}\;(25)\cdot \mathrm{Litter}\;\mathrm{size}$	(33)
$\mathrm{Daily}\;\mathrm{Allantoic}\;\mathrm{fluid}\;\mathrm{CP}\;\mathrm{deposition},\;g\cdot {\mathrm{day}}^{-1}=\mathrm{Equation}(26)\cdot \mathrm{Litter}\;\mathrm{size}$	(34)
$\mathrm{Daily}\;\mathrm{Placenta}\;\mathrm{CP}\;\mathrm{deposition},\;g\cdot {\mathrm{day}}^{-1\;}=\mathrm{Equation}\;(27)\cdot \mathrm{Litter}\;\mathrm{size}$	(35)
$\mathrm{Daily}\;\mathrm{Uterus}\;\mathrm{CP}\;\mathrm{deposition},\;g\cdot {\mathrm{day}}^{-1}=\mathrm{Equation}\;(28)\cdot \mathrm{Litter}\;\mathrm{size}$	(36)
$\mathrm{Daily}\;\mathrm{Fetus}\;\mathrm{CP}\;\mathrm{deposition},\;g\cdot {\mathrm{day}}^{-1}=\mathrm{Equation}\;(29)\cdot \mathrm{Litter}\;\mathrm{size}\cdot \frac{\mathrm{Piglet}\;\mathrm{birth}\;\mathrm{weight}\;}{1.46023}$	(37)
$\mathrm{Daily}\;\mathrm{Mammary}\;\mathrm{gland}\;\mathrm{CP}\;\mathrm{deposition},\;g\cdot {\mathrm{day}}^{-1}=\mathrm{Equation}\;(30)\cdot \;\mathrm{Available}\;\mathrm{teats}$	(38)

Input variables used in these equations: day: day of gestation; litter size: number of pigs born; piglet birth weight in g: average piglet birth weight in kg; available teats: number of available teats per gilt; body weight at breeding in kg.

**Table 7 animals-15-02126-t007:** Equations used in the daily amino acid (AA) deposition model.

$\mathrm{Daily}\;\mathrm{amniotic}\;\mathrm{fluid}\;\mathrm{wet}\;\mathrm{weight}\;\mathrm{deposition},\;g\cdot {\mathrm{fetal}\;\mathrm{pig}}^{-1}\cdot {\mathrm{day}}^{-1}=\mathrm{Equation}\;(1)\;(\mathrm{day}+1)-\mathrm{Equation}\;(1)\;(\mathrm{day});\;\mathrm{for}\;\mathrm{day}=1\;\mathrm{to}\;114$	(39)
$\mathrm{Daily}\;\mathrm{allantoic}\;\mathrm{fluid}\;\mathrm{wet}\;\mathrm{weight}\;\mathrm{deposition},\;g\cdot {\mathrm{fetal}\;\mathrm{pig}}^{-1}\cdot {\mathrm{day}}^{-1\;}=\mathrm{Equation}\;(4)\;(\mathrm{day}+1)-\mathrm{Equation}\;(4)\;(\mathrm{day});\;\mathrm{for}\;\mathrm{day}=1\;\mathrm{to}\;114$	(40)
$\mathrm{Daily}\;\mathrm{essential}\;\mathrm{AA}\;\mathrm{deposition}\;\mathrm{in}\;\mathrm{amniotic}\;\mathrm{fluid},\;g\cdot {\mathrm{day}}^{-1}=\mathrm{Equation}\;(39)\cdot \mathrm{Litter}\;\mathrm{size}\cdot \;\mathrm{Essential}\;\mathrm{AA}\;\mathrm{concentration}/1000$	(41)
$\mathrm{Daily}\;\mathrm{essential}\;\mathrm{AA}\;\mathrm{deposition}\;\mathrm{in}\;\mathrm{allantoic}\;\mathrm{fluid},\;g\cdot {\mathrm{day}}^{-1\;}=\mathrm{Equation}\;(40)\cdot \mathrm{Litter}\;\mathrm{size}\cdot \;\mathrm{Essential}\;\mathrm{AA}\;\mathrm{concentration}/10$00	(42)
$\mathrm{Daily}\;\mathrm{essential}\;\mathrm{AA}\;\mathrm{deposition}\;\mathrm{in}\;\mathrm{Placenta},\;g\cdot {\mathrm{day}}^{-1}=\mathrm{Equation}\;(27)\cdot \mathrm{Litter}\;\mathrm{size}\cdot \mathrm{Equation}\;(38)/100$	(43)
$\mathrm{Daily}\;\mathrm{essential}\;\mathrm{AA}\;\mathrm{deposition}\;\mathrm{in}\;\mathrm{Uterus},\;g\cdot {\mathrm{day}}^{-1}=\mathrm{Equation}\;(28)\cdot \mathrm{Litter}\;\mathrm{size}\cdot \mathrm{Equation}\;(38)/100$	(44)
$\mathrm{Daily}\;\mathrm{essential}\;\mathrm{AA}\;\mathrm{deposition}\;\mathrm{in}\;\mathrm{Fetus},\;g\cdot {\mathrm{day}}^{-1}=\mathrm{Equation}\;(29)\cdot \mathrm{Litter}\;\mathrm{size}\cdot \frac{\mathrm{Piglet}\;\mathrm{birth}\;\mathrm{weight}\;}{1.46023}\cdot \;\mathrm{Equation}\;(38)/100$	(45)
$\mathrm{Daily}\;\mathrm{essential}\;\mathrm{AA}\;\mathrm{deposition}\;\mathrm{in}\;\mathrm{Mammary}\;\mathrm{gland},\;g\cdot {\mathrm{day}}^{-1}=\mathrm{Equation}\;(30)\cdot \;\mathrm{Available}\;\mathrm{teats}\cdot \;\mathrm{Essential}\;\mathrm{AA}\;\mathrm{concentration}/100$	(46)
$\mathrm{Daily}\;\mathrm{essential}\;\mathrm{AA}\;\mathrm{deposition}\;\mathrm{in}\;\mathrm{Maternal}\;\mathrm{body},\;g\cdot {\mathrm{day}}^{-1}=\mathrm{Equation}\;(32)\cdot \;\mathrm{Essential}\;\mathrm{AA}\;\mathrm{concentration}/100\;$	(47)
$\mathrm{Daily}\;\mathrm{essential}\;\mathrm{AA}\;\mathrm{deposition}\;\mathrm{in}\;\mathrm{the}\;\mathrm{whole}\;\mathrm{body},\;g\cdot {\mathrm{day}}^{-1}=\mathrm{Equation}\;(41)\;+\;\mathrm{Equation}\;(42)\;+\;\mathrm{Equation}\;(43)\;+\;\mathrm{Equation}\;(44)\;+\;\mathrm{Equation}\;(45)\;+\;\mathrm{Equation}\;(46)\;+\;\mathrm{Equation}\;(47)$	(48)

Input variables used in these equations: day: day of gestation; litter size: number of pigs born; piglet birth weight in g: average piglet birth weight in kg; available teats: number of available teats per gilt; body weight at breeding in kg.

## Data Availability

The original contributions presented in this study are included in the article and Appendix A. Further inquiries can be directed to the corresponding authors.

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
