# Peer review of "Characterizing the Dynamic Protein and Amino Acid Deposition in Tissues of Pregnant Gilts: Implications for Stage-Specific Nutritional Strategies"

_animals, 2025, doi:10.3390/ani15142126_

Round 1

Reviewer 1 Report

Comments and Suggestions for Authors

The article presents a significant amount of work on protein metabolism in sows during gestation. The authors use the NRC model (2012) as a reference, seeking to explore and improve some of its critical aspects. In fact, they use almost the same factorial approach to obtain the global accumulation or retention models (amniotic fluid, allantoic fluid, placenta, uterus, fetuses, mammary gland, and maternal body).

In my opinion, rather than a scientific article, this is a draft report for presentation and discussion by a panel of experts. The conclusions it draws have little practical application to pig farming and are aimed at discus gestational physiological aspects and suggest some specific research lines.

Presented as a research article, I believe it is highly questionable on some basic points of great relevance:

  • Collection data: All results are obtained from bibliographic information. Growth models are obtained from an initial pool of 754 studies, only 19 studies are eligible, and the models are obtained with only 5 studies. Similarly, for protein and amino acid deposition models, 12 and 9 studies are eligible, and one and three are used, respectively. The use of a single study to obtain protein deposition models (Miller et al., 2016), which is the core of the work, cannot be compared to the work carried out by the NRC (2012), which includes numerous studies conducted between 1965 and 2003. It is true that it is a study carried out with more current genetics, but it is only one study.
  • The growth models presented in Table 4 are curvilinear, with a very diverse nature depending on the fraction considered. They undoubtedly are the best mathematical fitting and reflect the distribution of the data, but they do not offer a clear physiological explanation or meaning for each fraction. The authors themselves state: "The curve development process involved the combination of various equations encompassing a wide range of families such as exponential, power, growth, dose-response, sigmoidal, and distribution family models."
  • Occasionally, the authors opt for controversial and unconvincing models considering the data, such is the case of the placental lysine concentration (Fig. 4) or the whole protein deposition (Fig. 14).
  • The most important conclusion, in front of the NRC (2012), is probably that current genetics retain a greater amount of protein, especially at the beginning of gestation; although this may be true, it is not completely proven given that in the work of Miller et al (2016), the first nitrogen balance is performed on day 36 of gestation (see fig 15).

In short, this paper draft offers an interesting and thoughtful intellectual exercise, with a limited body of information (especially regarding protein deposition) that barely improves on the conclusions and recommendations of the NRC model (2012). In my opinion, this draft should be rejected for publication in Animal's.

Author Response

Reviewer 1

Comment 1: The article presents a significant amount of work on protein metabolism in sows during gestation. The authors use the NRC model (2012) as a reference, seeking to explore and improve some of its critical aspects. In fact, they use almost the same factorial approach to obtain the global accumulation or retention models (amniotic fluid, allantoic fluid, placenta, uterus, fetuses, mammary gland, and maternal body).

In my opinion, rather than a scientific article, this is a draft report for presentation and discussion by a panel of experts. The conclusions it draws have little practical application to pig farming and are aimed at discus gestational physiological aspects and suggest some specific research lines.

Response 1: We agree with the reviewer that the purpose of this study is to stimulate expert discussion rather than provide direct practical applications to pig farming. As stated throughout the manuscript, this work was not conceived as a requirement study. Instead, it seeks to generate hypotheses and, as the reviewer notes, improve critical aspects of previous models—particularly those used in the NRC (2012). We also agree that the conclusions primarily address physiological aspects of gestation and aim to identify specific research directions. The study follows the principles of Exploratory Data Analysis, which align with these objectives. However, we respectfully disagree with the characterization of the manuscript as a draft report. While its aim is exploratory, it adheres to the structure and standards of a scientific publication.

Comment 2: Presented as a research article, I believe it is highly questionable on some basic points of great relevance:

Collection data: All results are obtained from bibliographic information. Growth models are obtained from an initial pool of 754 studies, only 19 studies are eligible, and the models are obtained with only 5 studies. Similarly, for protein and amino acid deposition models, 12 and 9 studies are eligible, and one and three are used, respectively. The use of a single study to obtain protein deposition models (Miller et al., 2016), which is the core of the work, cannot be compared to the work carried out by the NRC (2012), which includes numerous studies conducted between 1965 and 2003. It is true that it is a study carried out with more current genetics, but it is only one study.

Response 2: We acknowledge the reviewer’s concern regarding the reliance on a single study (Miller et al., 2016) for estimating protein deposition. However, this decision was made intentionally and is grounded in the principles of Best Evidence Synthesis (BES), as described in the methodology section. BES emphasizes the use of the highest-quality data, rather than aggregating all available studies regardless of their methodological rigor or contextual validity. In this case, Miller et al. (2016) was considered the highest-quality dataset available, offering a well-designed study based on modern genotypes and production conditions—key factors for accurately modeling contemporary sow physiology. We therefore view the use of a single, high-quality, recent study as a strength rather than a limitation, especially given the alignment of our approach with the BES framework.

Comment 3: The growth models presented in Table 4 are curvilinear, with a very diverse nature depending on the fraction considered. They undoubtedly are the best mathematical fitting and reflect the distribution of the data, but they do not offer a clear physiological explanation or meaning for each fraction. The authors themselves state: "The curve development process involved the combination of various equations encompassing a wide range of families such as exponential, power, growth, dose-response, sigmoidal, and distribution family models."

Response 3: We appreciate the reviewer’s observation and agree that the models presented in Table 4 are empirical in nature. Our aim was to accurately describe the observed growth patterns of each physiological fraction using flexible, data-driven nonlinear functions, rather than to provide mechanistic or physiological interpretations for each curve. As stated in the manuscript, the model development process involved testing a wide range of mathematical families—including exponential, power, sigmoidal, and distribution-based functions—to capture the diversity of growth behaviors across tissues. This approach aligns with the principles of Exploratory Data Analysis (EDA) and nonlinear empirical modeling, where the primary goal is to fit and describe the data structure as faithfully as possible.

We fully recognize that these growth trajectories likely reflect a complex interplay of physiological processes. However, uncovering the specific biological mechanisms behind each curve was beyond the scope of this study and would require additional targeted experimental work. Our intention was to offer a descriptive framework that can inform future hypothesis generation and more mechanistic modeling efforts.

We interpret the reviewer’s suggestion as advocating for a mechanistic or compartmental modeling approach, which is valuable but relies on specific assumptions about underlying physiological processes—assumptions that were not available or verifiable within the current dataset. Our methodology, by contrast, remains agnostic to biological mechanisms and instead seeks to highlight and quantify the nonlinearities present in current data as a first step toward deeper understanding.

We agree that future studies should aim to integrate empirical and mechanistic approaches, and we see this work as a foundational step toward such integration.

Comment 4: Occasionally, the authors opt for controversial and unconvincing models considering the data, such is the case of the placental lysine concentration (Fig. 4) or the whole protein deposition (Fig. 14).

Response 4: We appreciate the reviewer’s comment and agree that the models in Figures 4 and 14 may appear unconvincing or even controversial. However, this was intentional and serves a specific purpose within the framework of Exploratory Data Analysis.

In Figure 4, the model for placental lysine concentration was deliberately shown specifically because of its poor fit, as it demonstrates a critical point: aggregating datasets from different sources may obscure important biological variability or produce misleading trends. The inadequate performance of the combined model in Figure 4 illustrates why a Best Evidence Synthesis approach was adopted in this study—prioritizing quality and contextual relevance over sheer quantity of data.

Similarly, the modeling choices in Figure 14 were designed to provoke critical evaluation. This figure exemplifies how the selection of a mathematical function, even with the same dataset, can yield divergent biological interpretations. While this may seem controversial, it highlights the dependency of conclusions on model structure and underlines the importance of careful model selection and transparency. These findings are not presented as definitive answers but rather as hypothesis-generating tools intended to guide expert discussion and future research.

Therefore, we do not view these controversial and unconvincing Figures as weaknesses. On the contrary, they are strengths of our approach. By showcasing model limitations and variability, we align with the core principle of EDA: to explore patterns, question assumptions, and encourage further inquiry rather than prematurely confirm fixed conclusions.

Comment 5: The most important conclusion, in front of the NRC (2012), is probably that current genetics retain a greater amount of protein, especially at the beginning of gestation; although this may be true, it is not completely proven given that in the work of Miller et al (2016), the first nitrogen balance is performed on day 36 of gestation (see fig 15).

Response 5: We agree with the reviewer’s observation that one of the key takeaways from this study is the hypothesis that current genetics may retain more protein during early gestation compared to the assumptions in NRC (2012). However, we respectfully clarify that this point was not presented as a definitive conclusion, but rather as a hypothesis and model-based prediction to guide future research.

As the reviewer notes, the data from Miller et al. (2016) begins at day 36 of gestation, which indeed limits the ability to confirm protein retention dynamics prior to that time point. We explicitly acknowledge this limitation and frame our findings accordingly. As stated in the conclusions:

“These predictions can serve as a foundation for in vivo studies aimed at optimizing nutritional strategies that enhance both the performance and well-being of swine during pregnancy.”

The objective of this work was not to confirm findings but to highlight emerging patterns based on the best available evidence, and to generate hypotheses for future empirical testing. In that context, we believe the distinction between prediction and confirmation was made clear throughout the manuscript.

Comment 6: In short, this paper draft offers an interesting and thoughtful intellectual exercise, with a limited body of information (especially regarding protein deposition) that barely improves on the conclusions and recommendations of the NRC model (2012). In my opinion, this draft should be rejected for publication in Animal's.

Response 6: We appreciate the reviewer’s perspective and agree that this manuscript does not aim to directly improve upon the dietary recommendations provided by the NRC (2012). However, we do not view this as a limitation, but rather as a reflection of the distinct objectives of the two works.

The NRC model was developed to generate practical feeding recommendations by aggregating data collected over several decades. In contrast, our study was not designed to formulate nutritional guidelines, but rather to conduct an exploratory, data-driven analysis of protein and amino acid deposition patterns in gestating sows. The purpose was to identify knowledge gaps, question assumptions, and generate hypotheses that can guide future experimental and mechanistic research. These objectives are clearly stated throughout the manuscript.

The absence of new dietary recommendations does not diminish the scientific contribution of this work. Instead, we view our study as complementary to the NRC (2012), offering a more contemporary and critical lens on nutrient deposition that reflects the physiology of modern swine genetics. We respectfully submit that this manuscript represents a constructive and necessary step toward improving future nutritional models and addressing the limitations of historical data sources.

Reviewer 2 Report

Comments and Suggestions for Authors

The article “Characterizing the dynamic protein and amino acid deposition in tissues of pregnant gilts: Implications for stage-specific nutritional strategies” is interesting, and the topic is important

As the authors explained, they built the model under the premise “for each tissue, a single study was chosen based on the Best Evidence Synthesis (BES) principle, which emphasizes using the best available dataset without aggregating all available studies.”

This method involves a possible bias in the output of the model. The authors mentioned it in the text, arguing that the huge variability could affect the model's inputs, and included Figure 4 to show the argument. 

With the figure data, we can see that if the authors used the data from the Jang paper, the model outputs should be pretty different. By the way, the text on lines 328 and 329 does not correspond to the Figure legend; one of them seems wrong.

I understand the authors' argument, but they should give the particular reason behind their choice in each case instead of a generic explanation as they did. 

In lines 375 to 382, the authors explain the incongruence in values obtained by the model, respecting empirical data, and include another equation to correct this problem. Thereafter, they conclude that an amino acid deficiency causes this situation. My first observation of this concern is that the three papers used to parametrize the model, as the authors said (lines 385 to 389), share the same feeder system, and that could explain the low feed intake, but they hypothesize a deficiency in amino acids. However, some papers conclude that overfeeding during that period could reduce embrionary survivance, a contrary conclusion achieved by the authors. On the other hand, another paper with the same conclusion with the difference that the other paper recommended an improvement of 31% of lysine respecting NRC (DOI: 10.3390/ani14131858).

Author Response

Reviewer 2

Comment 1: The article “Characterizing the dynamic protein and amino acid deposition in tissues of pregnant gilts: Implications for stage-specific nutritional strategies” is interesting, and the topic is important

As the authors explained, they built the model under the premise “for each tissue, a single study was chosen based on the Best Evidence Synthesis (BES) principle, which emphasizes using the best available dataset without aggregating all available studies.”

This method involves a possible bias in the output of the model. The authors mentioned it in the text, arguing that the huge variability could affect the model's inputs, and included Figure 4 to show the argument. 

With the figure data, we can see that if the authors used the data from the Jang paper, the model outputs should be pretty different. By the way, the text on lines 328 and 329 does not correspond to the Figure legend; one of them seems wrong.

Response 1: Thank you for pointing this out. We have corrected the text in lines 328 and 329 to ensure consistency with the figure legend. Specifically, the sentence:

“Jang et al. [3] reported Lys concentrations nearing 8 mg/g tissue, while Wu et al. [14] reported values up to 4 mg/g tissue.”

has been revised to:

“Jang et al. (2017)[3] reported Lys concentrations nearing 8 mg/g tissue, while Wu et al. (2017)[14] reported values up to 4 mg/g tissue.”

This correction ensures alignment between the in-text citation and the figure legend, clarifying the data sources used in Figure 4.

Comment 2: I understand the authors' argument, but they should give the particular reason behind their choice in each case instead of a generic explanation as they did. 

Response 2: We appreciate the reviewer’s comment and understand the request for more detailed, case-specific justifications. However, the selection of individual datasets for estimating wet weight, crude protein, and essential amino acid composition was conducted using a uniform set of criteria, as outlined in the methodology section.

Rather than making case-by-case decisions, we applied the same objective selection process across all tissues and components to ensure methodological consistency and minimize selection bias. As a result, we are unable to provide specific justifications for each dataset beyond the established criteria, as no subjective or contextual decision-making occurred at the individual case level.

The example shown in Figure 4 was intended to illustrate the broader rationale for applying the Best Evidence Synthesis (BES) approach—namely, to demonstrate how aggregating heterogeneous datasets can produce misleading representations of physiological trends. This example was not part of the actual selection process and was used solely to justify the overarching strategy. As stated in the results section:

“Due to the significant differences among the reported values, the BES principle was applied. This ensured that tissue growth and development were modeled using the highest-quality dataset, rather than aggregating conflicting data.”

We hope this clarifies that our dataset selections were based on a systematic, reproducible framework, rather than individual or subjective decisions.

Comment 3: In lines 375 to 382, the authors explain the incongruence in values obtained by the model, respecting empirical data, and include another equation to correct this problem. Thereafter, they conclude that an amino acid deficiency causes this situation. My first observation of this concern is that the three papers used to parametrize the model, as the authors said (lines 385 to 389), share the same feeder system, and that could explain the low feed intake, but they hypothesize a deficiency in amino acids. However, some papers conclude that overfeeding during that period could reduce embrionary survivance, a contrary conclusion achieved by the authors. On the other hand, another paper with the same conclusion with the difference that the other paper recommended an improvement of 31% of lysine respecting NRC (DOI: 10.3390/ani14131858).

Response 3: We appreciate the reviewer’s detailed observations. While it is true that three of the studies used to parameterize the model shared the same feeder system, a fourth study (lines 387–389) involved animals housed in stalls without that system and still show the same early gestation weight loss, suggesting that feeder type alone may not explain the phenomenon. As we noted in line 391, “This intriguing consistency across different studies warrants further investigation to fully understand its underlying causes.” The hypothesis of a potential amino acid deficiency was formulated on this basis, not as a definitive conclusion but as a starting point for further research.

We acknowledge that some studies have reported that overfeeding during early gestation may impair embryonic survival. However, “overfeeding” is a subjective term, often dependent on the reference baseline. For instance, a lysine intake of 17 g SID/d during early gestation may be seen as excessive under NRC (2012) guidelines but is consistent with standard practice under European feeding recommendations. To reduce this subjectivity, we referenced aggregated evidence, such as the review by Langendijk [54], cited in line 530, which supports increased AA intake in early gestation for improved luteal function and embryo survival.

Importantly, we do not attribute the observed inconsistency solely to amino acid deficiency. As discussed in line 644, “Pregnant gilts may require increased levels of specific AAs and/or protein, along with additional energy during early gestation…” suggesting that both nutrient profile and energy balance may be involved.

This study was conducted under the framework of Exploratory Data Analysis, which aims to generate hypotheses rather than confirm established conclusions. As stated in the conclusion:

“These predictions can serve as a foundation for in vivo studies aimed at optimizing nutritional strategies that enhance both the performance and well-being of swine during pregnancy.”

We hope this work provides mechanistic insights and a data-driven foundation to help clarify conflicting findings—such as the one highlighted by the reviewer—and to guide future experimental research.

Reviewer 3 Report

Comments and Suggestions for Authors

This article is impressive and sheds interesting light on pig nutrition.

I have only one observation: the results may have been impacted by the heterogeneity of the articles in terms of timing. Indeed, given the evolution of pig genetics, can we compare a sow from 1977 with one from 2024? But this is a flaw inherent to this type of holistic analysis and is unfortunately unavoidable.

Author Response

Comment 1: I have only one observation: the results may have been impacted by the heterogeneity of the articles in terms of timing. Indeed, given the evolution of pig genetics, can we compare a sow from 1977 with one from 2024? But this is a flaw inherent to this type of holistic analysis and is unfortunately unavoidable.

Response 1: We thank the reviewer for this thoughtful observation. We agree that the heterogeneity in the timing of the source articles—particularly in relation to evolving pig genetics—poses a limitation. However, we also see this not only as a constraint but as a key result of the analysis itself: it underscores the lack of contemporary, high-quality data needed to support holistic modeling approaches like the one presented in this manuscript. This gap in the literature reinforces the need for renewed data collection efforts using modern genotypes and production systems, which we believe our work helps to highlight.

Reviewer 4 Report

Comments and Suggestions for Authors

The manuscript entitled “Characterizing the dynamic protein and amino acid deposition in tissues of pregnant gilts: Implications for stage-specific nutritional strategies” developed a gestation model that characterizes the dynamic changes in growth, crude protein, and amino acid deposition throughout gestation. The outcomes have a great significance for  precision feeding strategies that enhance pig health, reproductive outcomes, and farm efficiency. The models built in the study provide a more detailed perspective on protein and amino acids deposition throughout gestation. These predictions can serve as a foundation for in vivo studies aimed at optimizing nutritional strategies that enhance both the performance and well-being of swine during pregnancy. There are a few comments listed as following:

Comments

  • Thebasis for the selection of the data included in themodel construction needs to be further explained. Such as Line 96-97 & 106-107.
  • Themethods of data analysis and model building are recommended to be placed in the part of “Statistical Analysis”.
  • How to determine the accuracy of the model? Only R2? The basis for judgment and data should also be indicated in the Figures and Tables.

Author Response

Comment 1: The basis for the selection of the data included in the model construction needs to be further explained. Such as Line 96-97 & 106-107.

Response 1: We thank the reviewer for this helpful comment and agree that the criteria for study selection required further clarification. To address this, we have revised the relevant text to more clearly explain the rationale behind the inclusion of specific studies for model construction.

Specifically:

Original: “The selected studies were those from the eligible pool with the most time points for wet weight and CP content, as well as those that covered the longest gestational period, which was the strongest criterion in the selection process.”

Revised to: “The selected studies from the eligible pool were those that reported the most time points for wet weight and CP content, and especially those that covered the longest duration of the gestational period—this was considered the strongest criterion in the selection process. This approach prioritized broader data coverage across gestation, enabling a more accurate and continuous representation of temporal growth and deposition patterns. When two studies reported a similar number of time points, the one with broader gestational coverage was selected.”

---

Original: “Following the BES principle, the selected study was chosen from the eligible pool based on having the most time points and covering the longest gestational period.”

Revised to: “Following the BES principle, the selected study was chosen from the eligible pool based on the number of time points and the duration of gestational coverage, with preference given to studies offering broader data coverage when both criteria were similar.”

---

Original: “The selected studies for each tissue were those from the eligible pool that included the most time points and reported all essential AA.”

Revised to: “The selected studies for each tissue were those from the eligible pool that included the most time points and reported all essential amino acids. When studies had a similar number of time points, preference was given to those with broader gestational coverage.”

---

These revisions aim to improve transparency and align with the reviewer’s request by making the selection criteria more explicit. We hope this resolves the concern.

Comment 2: The methods of data analysis and model building are recommended to be placed in the part of “Statistical Analysis”.

Response 2: We appreciate the reviewer’s suggestion. However, we would like to clarify that the modeling approach used in this study does not align with conventional statistical hypothesis testing frameworks that are typically described in a “Statistical Analysis” section. Instead, we developed a mechanistic, data-driven model that integrates multiple tissue components through an algorithmic process based on physiological structure, rather than through inferential statistical comparisons.

While some statistical tools—such as R-squared values and polynomial curve fitting (e.g., in Figure 14)—were used during post-hoc evaluations of model predictions, these were not employed for hypothesis testing or significance inference. Rather, they were used for data visualization and model interpretation as part of the broader discussion. These tools helped illustrate how different modeling choices could yield varied interpretations, a central aspect of our Exploratory Data Analysis (EDA) framework.

Because the aim of this work is to generate hypotheses, not confirm them through traditional statistical inference, we believe including a “Statistical Analysis” section might mislead readers into assuming that confirmatory statistical tests were used to support definitive conclusions. Therefore, to maintain alignment with the exploratory and hypothesis-generating nature of the study, we have chosen not to include a standalone statistical analysis section.

We hope this explanation clarifies our rationale and aligns with the broader methodological principles guiding this work.

Comment 3: How to determine the accuracy of the model? Only R2? The basis for judgment and data should also be indicated in the Figures and Tables.

Response 3: The accuracy of the model was assessed primarily based on its biological plausibility and internal consistency, rather than on statistical fit alone. While R² values indicated a strong statistical agreement between model predictions and observed data, we recognize that relying solely on mathematical error metrics—such as R² or others—can be misleading in this context. This is because the model evaluation was conducted using aggregated mean values from selected studies rather than individual animal-level data, which can artificially inflate goodness-of-fit metrics and obscure underlying variability.

Therefore, we emphasized biological relevance as a more meaningful criterion for assessing model accuracy. Specifically, we examined whether the predicted growth trajectories aligned with known physiological patterns in pregnant gilts. For example, as noted in the manuscript, “the predicted whole-body weight gain during gestation for Buis [46], Thomas et al. [47], and Stewart [48] was 73.1, 76.8, and 72.8 kg, respectively. In contrast, empirically reported values are approximately 62 kg [1].” This discrepancy prompted a recalibration of the model in Section 3.1.1, despite its strong statistical fit.

We made a deliberate decision not to overemphasize mathematical performance metrics for two key reasons:

  1. To avoid overstating the certainty of our estimates. High goodness-of-fit metrics could falsely suggest that the model outputs represent confirmed biological facts, whereas the study's aim was to explore potential growth and deposition patterns and generate hypotheses.
  2. Because standard performance metrics applied to means are not methodologically appropriate in this case. Aggregated mean values reduce within-group variance and can mask important biological variation, leading to inflated fit statistics that do not reflect true prediction accuracy.

In response to the reviewer’s comment, we have added the following clarification in Section 3.1.1 to address this point directly:

“Thus, although R² values indicate a strong statistical fit, goodness-of-fit metrics were not the primary evaluation criterion for the model. Instead, the main emphasis was placed on biological relevance—that is, whether the predicted outputs aligned with known physiological patterns in pregnant gilts. This biologically grounded evaluation was considered more appropriate given the model’s exploratory purpose and the use of aggregated data (Figure 5), which can artificially inflate statistical fit.”

Ultimately, the goal of the model was not to establish a definitive growth equation, but to evaluate the biological coherence of protein and amino acid deposition calculations across gestation. For this reason, although R² values were reported as indicators of statistical fit, they were not emphasized—and are therefore not presented in the figures or tables. Instead, biological interpretation was prioritized, as the model was designed as an exploratory tool to identify knowledge gaps, test assumptions, and support future investigations in gestating gilt nutrition.

Round 2

Reviewer 1 Report

Comments and Suggestions for Authors

No comments or sugestions for the authors